# Development and Research of Recyclable Composite Metamaterial Structures Made of Plastic and Rubber Waste to Reduce Indoor Noise and Reverberation

**Andrej Naimušin** *[ID] **and Tomas Januševičius** [ID]

Department of Environmental Protection and Water Engineering, Vilnius Gediminas Technical University, 10223 Vilnius, Lithuania
*   Correspondence: andrej.naimusin@vilniustech.lt

**Abstract:** According to the Waste Management Policy of the European Union, the recycling and reuse of various wastes is considered the most ecological and advanced waste disposal technology with the lowest impact on the environment. By applying circular economy principles, plastic waste will extend its life cycle and be used as secondary raw materials to create structures with good sound insulation and absorption properties. Structures created from metamaterial with plastic were studied for their sound-absorbing properties in an impedance tube. A combined 100 mm long resonator design with a 2.0 mm perforation, 20 mm thick plate, regardless of whether it is an "X"- or "O"-shaped resonator, achieved a good sound absorption peak at 315 Hz of 0.94. When the combined structures of 50 mm long "X"- and "O"-shaped designs were compared, different sound absorption peaks at high frequencies were achieved. A slightly better sound absorption peak of 0.95 was achieved at 500 Hz with the "O"-shaped resonator and 0.93 at 630 Hz with the "X"-shaped resonator. The results show that the combined plastic and rubber structures can be integrated into building structures and be used as an alternative to improve building acoustics and reduce noise and reverberation.

**Keywords:** sustainability; green construction; building design; sound insulation; sound absorption; plastic waste; rubber waste; acoustic metamaterial; resonator





## 1. Introduction

Plastic is one of the most important elements of modern life and is widely used in all areas. Every year more than 300 million tons of plastic materials are produced, most of which are thrown into the ocean, buried in landfills, or simply burned. Plastic waste causes serious global environmental and health problems such as pollution, loss of biodiversity, energy, and economic losses [1]. The accumulation of plastic waste and the lack of proper disposal methods has created a critical and unprecedented problem where plastic waste enters our water resources and waterways, overflows landfills, leaches into the soil, and enters the air, polluting all natural objects and resources in our environment [2].

Currently, as the number of cars in the world increases, the number of worn tires, the disposal problems of which have not been solved, is constantly increasing. According to "Eurostat", more than 5 million tons of unusable vehicle waste is generated annually in EU countries, and about 88% of this amount is reused or can be recycled [3].

About 20% of tires are recycled through retreading, a small proportion is recycled for energy, and some are shredded as secondary raw material to create new products. However, a large proportion of used tires still end up as waste. A circular economy product consumption model that involves sharing, renting, reusing, repairing, refurbishing, and recycling the material as long as possible could be used in this situation. In this way, the life cycle of the products is extended. A comprehensive review of the literature on the circular economy can be found in Suzanne et al. (2022) [4], as well as in the work of Acerbi and Taisch (2022) [5]. The issue of the circular economy related to the tire recycling problem

has been published in a study by Gigli et al. (2019) [6] and the automotive industry by Agyemang et al. (2019) [7] and Vermesan et al. (2019) [8].

However, there is a high potential for using recycled rubber (rubber granules) in the production of noise reduction panels to improve the sound insulation and absorption properties of building elements, or as an environmental noise reduction barrier to improve the quality of the urbanized environment.

This research envisages the production of a new generation product—noise-reducing structures that are characterized by high sound absorption, are tolerant of moisture, and are made from secondary raw materials. One such material is plastic, the secondary use of which is an issue of concern for the whole world. The recycling of worn tires has to date not been attempted by grinding up the remaining synthetics that are expected to be used in this research. A 3D printing technology is used to print a perforated composite structure with resonators, the purpose of which is to isolate and absorb incoming sound. For the absorbent layer, we propose the use of tire synthetics, represented by mineral wool, which, when combined with a plastic resonator, should have good absorbent properties. Structures made of plastic are designed and created using 3D printing technologies, and absorption and insulation properties are tested with an interferometer. The acoustic properties of synthetic cord and the absorption and sound insulation characteristics of composites with synthetic cord fiber and plastic are investigated. Based on the research data received, the most optimal options are chosen.

## 2. Literature Review

This chapter presents an analysis of the literature on the generation and secondary use of plastic and rubber waste, the use of acoustic metamaterials, and resonators.

### 2.1. Generation and Secondary Use of Plastic Waste

Recently, the waste management sector has been increasingly applying the circular economy concept in order to reduce the amount of waste generated and the negative impact on the environment. There are many sectors where we can use or recycle plastic waste for further use, such as building materials, turning plastic waste into fuel, household items, fabrics and clothing, shoe soles, etc. Figure 1 presents the latest packaging waste statistics for the 27 member states of the European Union (EU) and some non-EU countries. The data presented summarize the changes between 2009 and 2020, when official reports on packaging waste were implemented in all EU member states. The information and data are based on Directive 94/62/EC, which defines the purposes of processing and use. This directive aims to ensure a high level of environmental protection and harmonize national packaging and packaging waste management measures.

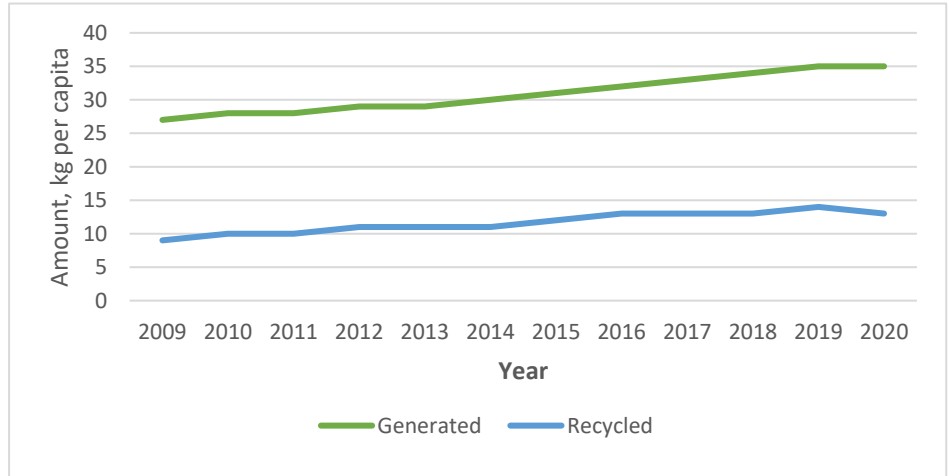

**Figure 1.** Amounts of plastic waste generated and processed in the EU 2009–2020, kg per capita [3].

Such large amounts of waste generation and recycling show the relevance of research into the use of secondary raw materials and the search for applying them in other spheres, for example, improving the interior acoustics of buildings, improving insulation, and/or room reverberation.

There are different ways to treat and recycle plastic waste. The first step in processing is to shred the waste to the desired size, which will allow it to be further processed by using more advanced recycling processes [9]. In 2016, R. Dharmaraj et al. used a metal pestle to grind low-density polyethylene (LDPE) melted at 120 °C into a semiliquid form. The remaining particles are then sieved through a 4.75 mm sieve. The tests were carried out using a rotary grinder to grind the waste [10]. However, simpler solutions are also being discovered, one of which was described in 2015 by Chen et al. by directly shredding plastic waste in a screw extruder. After pre-treatment, the plastic pellets are sent back to the extruder together with the filler and binder. In this process, the size of the plastic particles does not matter because, when the plastic melts, the different segments stick together to form a single material. Evenly spreading plastic throughout the material gives a good base to the product, in this case, bricks [11].

Today, increasing consumer awareness has started to encourage people to use natural or recycled materials, which has increased the search for these materials in sectors such as construction and acoustics, and more specifically as sound insulation and absorption materials. The construction sector plays a very important role in finding possible uses for renewable resources. D'Amore et al. reported an innovative method for the production of glass foams. This glass foam could be a viable alternative to rock wool insulation, as it works well as a sound insulator, reducing the footprint, weight, and costs associated with $CO_2$ emissions while using very little waste for production. The percentage of glass used affects the final density of the styrofoam. Acoustic tests and scanning electron microscopy have shown an open cell structure that improves sound absorption in the mid-frequency range and also improves thermal insulation properties [12].

Another example is the research by Hande Sezgin et al., who tried to design high-value-added composite panels that act as an alternative thermal and acoustic material by combining textile and packaging waste from two different sectors. The use of both textile and packaging waste as composite components allows the production of 100% recyclable insulation panels, which is the new method of manufacturing composite insulation panels. The results showed that the properties of heat and sound insulation of the porous composite panels made from packaging and textile waste were comparable to those of usual commercial building materials. The most important physical property that controls air sound transmission loss through a material is mass per unit area. Therefore, when studying 5 mm thick panels mixed with polypropylene or polyethylene, the researchers obtained a sound transmission loss of 6 dB and 8 dB at 1000 Hz, and when studying 20 mm–2 and 3 dB [1].

Several researchers have attempted to use polypropylene (PP) fiber waste to produce functional fabrics or sound-absorbing and thermal-insulating materials. For example, Lin et al. produced impermeable acoustic and thermal insulation composite materials using recycled Kevlar, nylon, and low-melting polyester nonwovens, which were reinforced with recycled polypropylene nonwoven seams. In this study, a non-woven composite was produced by inserting PP non-woven yarns in different directions between Kevlar, nylon, and low-melting polyester non-woven fabrics. The addition of PP waste was found to increase the sound absorption of the composites by 0.2 at 2224 Hz. The five-layer composite materials of the authors had the lowest thermal conductivity of 0.047 W/mK and a sound absorption coefficient of more than 0.94 at frequencies above 1890 Hz [13].

Recently, Jamnongkan et al. showed that blending recycled polypropylene (rPP) using a melt-extrusion process can be added into the polypropylene (PP) matrix without significantly affecting the mechanical properties. All prepared samples showed good performance of mechanical properties. However, the modulus and tensile strength of PP slightly decreased with increased rPP concentrations. This tendency could be a result of

a melt-extrusion process itself. Researchers added carbon black to improve the electrical conductivity of recycled PP. Comparative results of PP, rPP at various concentrations, and rPP with carbon black showed that it may be possible to develop a technology for PP waste, and with value-added materials in mind even improve some of its properties [14].

Plastic recycling is highlighted as an important step in the transition to a circular economy to avoid the use of fossil resources and close the plastic recycling cycle. The concept (Figure 2) is that we take plastic waste of various types and used tires as raw materials and make secondary materials from them. Sorted plastic waste can be melted into desired shapes or used to make 3D printing filler if we want to make something very specific with precise geometries (metamaterial or a resonator). Rubber tire waste can be technologically recycled (shredded) and made into rubber granules, which can be used as an absorbing material in building construction. At the end of life, we can collect those building structures and parts and recycle them in order to extend their life cycle. Recycled materials can be used again to make new products for use in building construction.

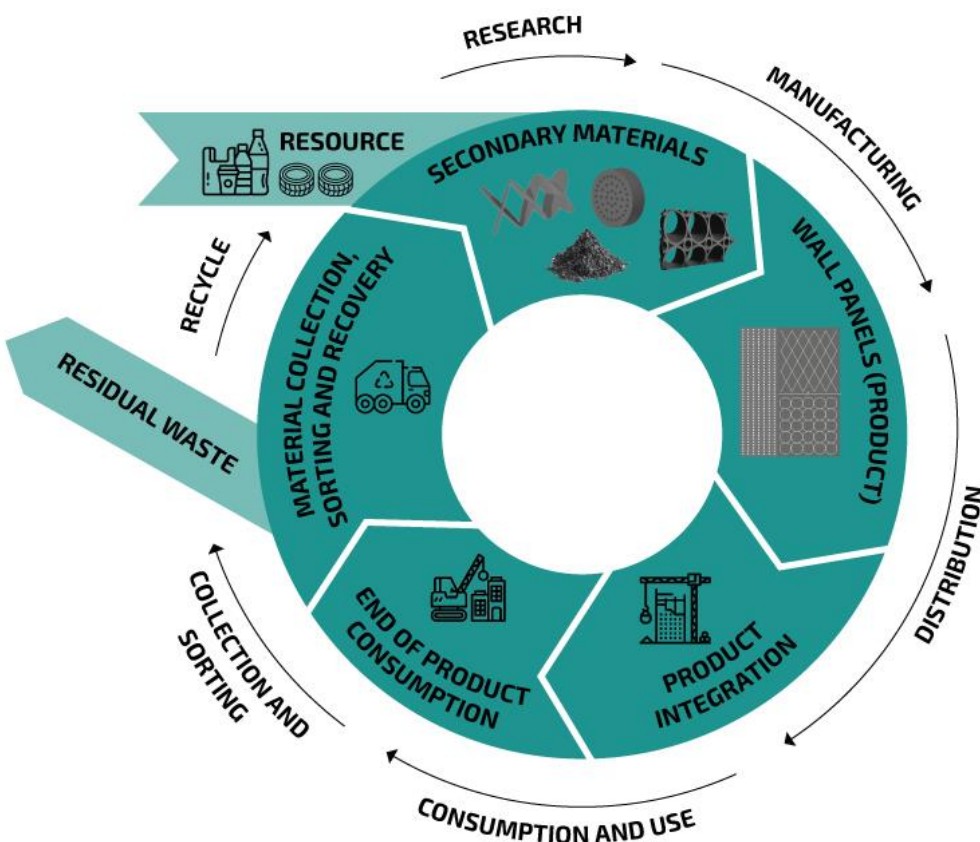

**Figure 2.** Plastic and rubber waste circular economy concept in building construction.

In order to make optimal and better solutions to the use of recycled secondary materials, we need to research fields of interest. A common theme of these studies is bulk density, thickness, porosity, and tortuosity, the main factors influencing the heat transfer, sound absorption, and insulation properties of a new material.

### 2.2. Generation and Secondary Use of Rubber Waste

The most common and widely used utilization of tire waste according to the principles of the circular economy and the waste hierarchy is granulation to obtain rubber granules and rubber powder. However, there is a high potential for using recycled rubber (rubber granules) in the production of noise reduction panels to improve sound insulation and absorption properties of building elements, or as an environmental noise reduction barrier to improve the quality of the urbanized environment.

To solve the problem of huge waste, scientists began to study the sound absorption properties of various materials and evaluated their applicability potential from this point of view. For example, to make a barrier 1 km long and 3 m high, about 65.5 tons of rubber granules are used, which are obtained from 7800 recycled tires [15].

Pfretzschner and Rodriguez (1999) confirmed that rubber granules can be a good sound absorber with a broad-band absorption spectrum, which is an excellent alternative to the current sound-absorbing screens used to protect against traffic noise and, at the same time, remove waste tires from landfills [16]. Swift et al. (1999) found that this material can effectively absorb sound if the rubber granules and binder content are carefully selected and the thickness of the material is adjusted to the frequency range of interest [17]. Hong et al. (2007) showed that a new sound absorber with recycled rubber particles has good damping properties as a sound pressure damping layer. Researchers highlight low manufacturing cost, broadband sound absorption, thin thickness, and relatively simple processing as advantages [18].

Acoustic materials play an important role in acoustic engineering in reducing noise levels in various applications. Typically, sound absorption material is used to overcome the unwanted effect of sound reflection and also helps to reduce the level of return noise. The thickness of the absorbent material is one of the most important parameters that affect acoustic properties. The best sound absorption factor depends on the density of the material, which is an important parameter to consider in any study.

### 2.3. Use of an Acoustic Metamaterial with Resonators

A metamaterial is any artificial material created from microscopic arrangements of existing elements in a structure that gives the material unconventional properties, especially when used to manipulate light or sound waves [19].

These artificial structures have enabled new functions such as negative effective properties, extraordinary wave manipulation, better sound absorption and isolation, acoustic cloaking, acoustic wave focusing, and efficient sound wave energy harvesting [20]. As an example, based on the hybrid core of honeycomb and corrugation (K-C hybrid core), introducing perforations in both the upper surface layer and the corrugation, researchers Tang et al. in 2017 formed an acoustic metamaterial in order to obtain ultra-wideband low-frequency sound absorption. Using microperforated plate (MPP) theory, the researchers establish a theoretical method to calculate the sound absorption coefficient of this new type of metamaterial. The theoretical and numerical results showed good agreement, proving that this new 60 mm thick sound absorber can achieve perfect absorption at around 580 Hz with broadband absorption bandwidth. The strong energy dissipation due to the perforation is attributed to the Helmholtz resonance effect [21]. Other researchers have proposed a complex honeycomb acoustic metamaterial based on the characteristics of the honeycomb structure, the Helmholtz resonator, and acoustic metamaterials [22].

Acoustic metamaterials provide increasing opportunities to develop targeted acoustic properties through the development of functional materials. Properly designed acoustic metamaterials exhibit non-negative properties that are often difficult to realize from natural materials. The increased interest in acoustic metamaterials and the freedom of their design provided by computer-aided design manufacturing (for example, 3D printing) will allow the development of a new generation of acoustic metamaterials in the near future that are smaller and more efficient, allowing for expanded acoustic characteristics [23].

Plastic acoustic metamaterials are usually designed and manufactured with the help of 3D printing technologies, as this allows the creation of unique and reproducible materials with the desired acoustic parameters.

Acoustic resonators are commonly used in noise control engineering, where relatively simple, durable solutions are desired. Acoustic resonators typically absorb a very narrow band of frequencies but are often used in applications where the noise is constant and tonal. Although there are many types of acoustic resonators, the two most common are the Helmholtz resonator and the quarter-wave resonator [24].

A Helmholtz resonator is a type of resonator consisting of an empty cavity and a narrow neck. Nowadays, you can come across various variants of this device, in building acoustics, it is mostly individual resonators that are used or combinations of structures formed by perforated plates. Many researchers in the past have studied the behavior of Helmholtz resonators in different cases. The pioneer in this field was Ingard, who published a paper in 1953 describing the behavior of resonators with different neck and cavity shapes and defining the main design relationships of these resonators [25].

A subsequent study by Jena et al. was performed using eight different configurations of Helmholtz resonators, in parallel and series, using similar and dissimilar resonators. The performance of each configuration was investigated analytically, numerically, and experimentally validated. It has been observed from research that a single Helmholtz resonator can be considered an acoustic metamaterial with negative compressibility. However, both properties become negative when two similar or dissimilar resonators are arranged in parallel instead of in series in the direction of sound wave travel. In the case of a double Helmholtz resonator configuration, the effective properties become negative in two distinct zones corresponding to the two resonance frequencies of the resonator [26].

A quarter-wave resonator is simply a tube that is closed at one end and open at the other. The walls and the rigid end of the tube isolate air from entering the resonator cavity. Under certain conditions, this air accumulation can enter a resonant state. However, in engineering, they are usually located in industrial buildings to reduce the noise emitted by the fans of air-circulating systems, combined with a barrier, reducing ambient noise [27].

Although resonators and their principle of operation as individual systems are known and widely applied in acoustic engineering, their utility and integration into building structures is a relatively new field.

### 3. Methodology and Results

This section presents research graphs of measurement results, their description and summary, insights, and partial conclusions.

*3.1. Research Methodology*
Determination of the Acoustic Parameters of the Material Using an Impedance Tube

Sound absorption is characterized by a sound absorption coefficient, which ranges from 0.00 to 1.00, where 1.00 indicates perfect absorption of sound energy (no reflection) and 0.00 means that the material does not absorb sound incident on it and reflects all of its energy.

According to the European Committee for Standardization, there are two methods of determining the sound absorption coefficient using an impedance tube. The first is the standing wave ratio method and the second is the transfer function method. The transfer function method is similar to the first test method, in that it uses an impedance tube with a sound source (loudspeaker) connected at one end and a sample at the other end, but the measurement technique is different. In this test method, plane sound waves are generated in the tube by a sound source (loudspeaker) and the decomposition of the field is achieved by measuring the sound pressure at two fixed locations using wall microphones or a transverse microphone in the tube and then calculating a complex acoustic transfer function, and the normal frequency absorption and impedance coefficients of the acoustic material [28].

This method is based on the fact that the complex reflection coefficient $R$ can be determined by measuring the transfer function $H_{12}$ between two positions at distances $x_1$ and $x_2$ from the test sample.

Sound pressure measurement determines the pressure at each frequency:

$$H_{13} = \frac{p_3}{p_1}, \; H_{23} = \frac{p_3}{p_2}, \tag{1}$$

where $H_{13}$—the transfer function between microphones no. 1 and no. 3; $H_{23}$—the transfer function between microphones no. 2 and no. 3; $p_3$—the pressure recorded by the third microphone; $p_2$—the pressure recorded by the second microphone; $p_1$—the pressure recorded by the first microphone.

$$k_0 = \frac{2\pi f}{c_0}, \tag{2}$$

where $k_0$—the sound wave number in the air; $f$—the frequency; $c_0$—the speed of sound in air.

$$H_{I\ (160-1000\ \text{Hz})} = \frac{p_{3I}}{p_{1I}} = e^{-jk_0(x_{12}+x_{23})}; \ H_{I\ (1-5\ \text{kHz})} = \frac{p_{3I}}{p_{2I}} = e^{-jk_0(x_{23})}, \tag{3}$$

where $H_I$—the incident wave transfer function; $s$—the distance between the microphones; $k_0$—the sound wave number in the air; $j$—the index of a complex number; $x_{12}$—the distance between microphones no. 1 and no. 2; $x_{23}$—the distance between microphones no. 2 and no. 3.

$$H_{R\ (160-1000\ \text{Hz})} = \frac{p_{3R}}{p_{1R}} = e^{jk_0(x_{12}+x_{23})}; \ H_{R\ (1-5\ \text{kHz})} = \frac{p_{3R}}{p_{2R}} = e^{jk_0(x_{23})}; \tag{4}$$

where $H_R$—the transfer function of the reflected wave; $s$—the distance between the microphones; $k_0$—the sound wave number in the air; $j$—the index of a complex number; $x_{12}$—the distance between microphones no. 1 and no. 2; $x_{23}$—the distance between microphones no. 2 and no. 3.

The reflection coefficient ($R$) is calculated according to the formula:

$$R_{(160-1000\ \text{Hz})} = \frac{H_{13} - H_{I\ (160-1000\ Hz)}}{H_{R\ (160-1000\ Hz)} - H_{12}} e^{2jk_0(X_{12}+X_{23}+X_{3s})};$$
$$R_{(1-5\ \text{kHz})} = \frac{H_{23} - H_{I\ (1-5\ kHz)}}{H_{R\ (1-5\ kHz)} - H_{13}} e^{2jk_0(X_{23}+X_{3s})}, \tag{5}$$

where $H_{13}$—the transfer function between microphones no. 1 and no. 3; $H_{23}$—the transfer function between microphones no. 2 and no. 3; $H_I$—the incident wave transfer function; $H_R$—the transfer function of the reflected wave; $k_0$—the sound wave number in the air; $j$—the index of a complex number; $X_{3S}$—the distance between microphone no. 3 and sample; $x_{12}$—the distance between microphones no. 1 and no. 2; $x_{23}$—the distance between microphones no. 2 and no. 3.

From this we finally obtain the absorption coefficient for plane waves:

$$\alpha = 1 - |R|^2, \tag{6}$$

In further research studies of structures with plastic and rubber, the second method of determining the transfer function, the sound absorption coefficient with an impedance tube (Figure 3) will be applied because it is simpler and faster to determine the sound absorption coefficient and other relevant parameters for material characterization.

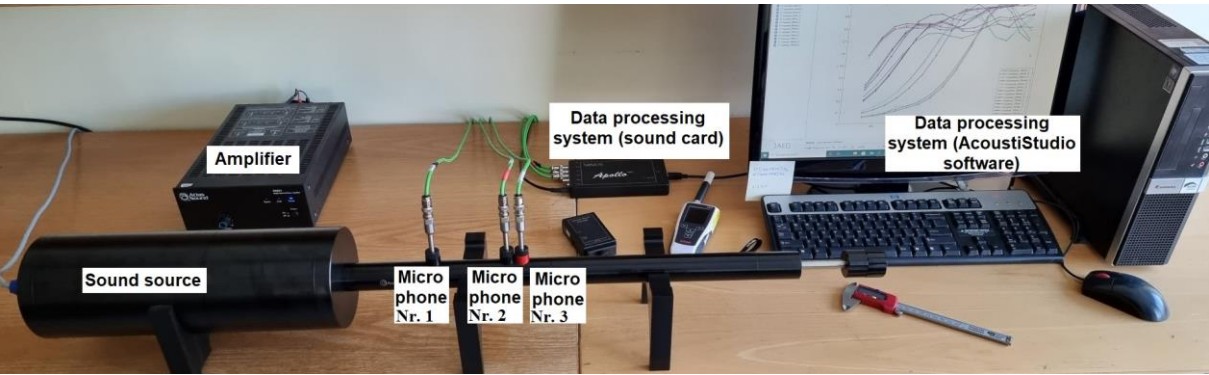

**Figure 3.** Arrangement of the interferometer and its components for the determination of sound absorption.

### 3.2. Sound Absorption Results of Composite Metamaterial Structures with Resonator Using Impedance Tube

The samples used are circular and about 29.0 mm in diameter to fit in the impedance tube. The sample is made up of three parts, these are plates of different thicknesses and perforation levels, different lengths and types of resonators, different amounts of filler from coarse fraction rubber granules (%), and different lengths of resonator and filler holders (Figure 4). The complete assembled structures and their individual parts were measured (Table 1) in order to obtain the full influence of each part of the structure on sound absorption and to draw the appropriate conclusions.

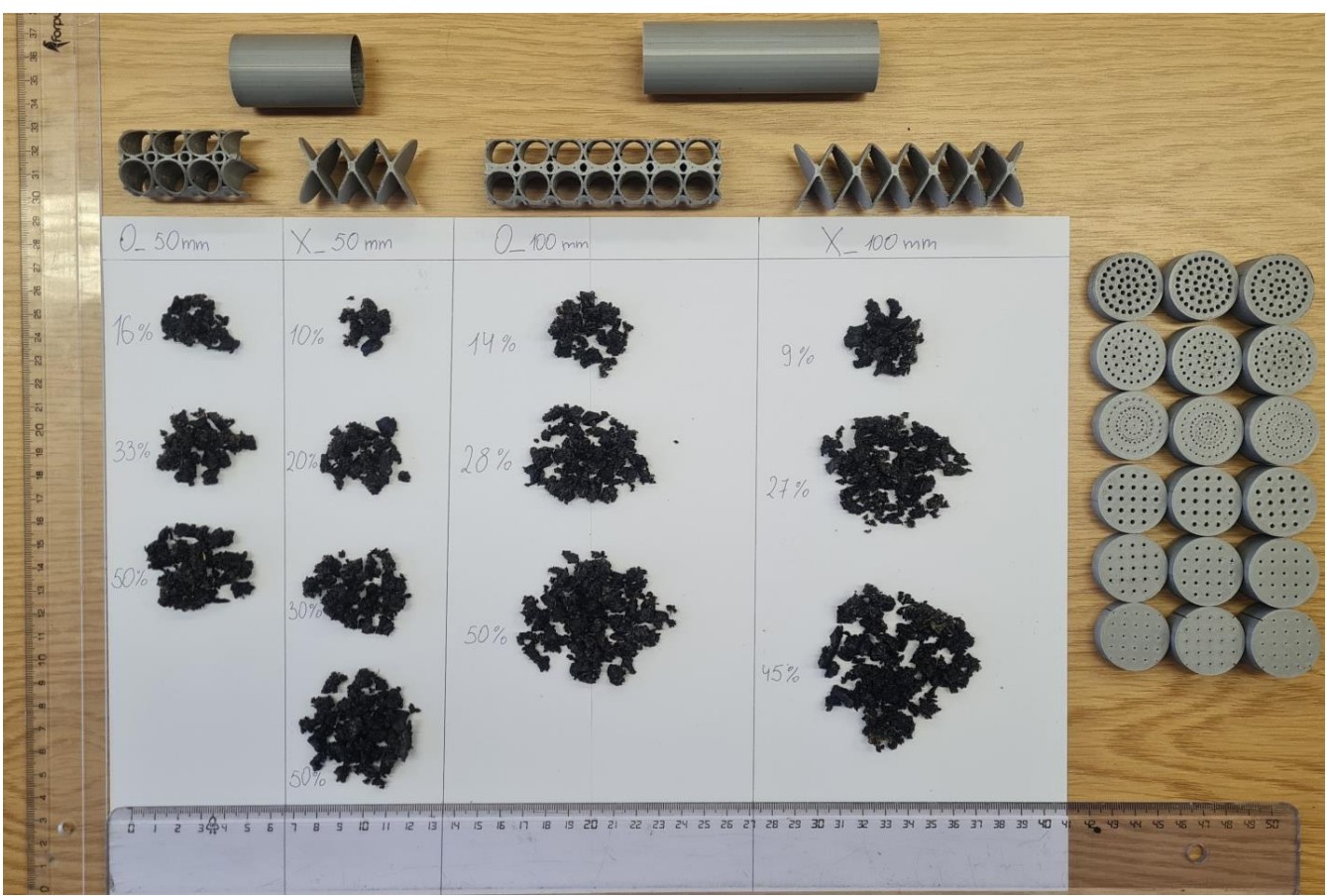

**Figure 4.** Plates of different thicknesses and perforation levels, different lengths and types of resonators, different amounts of filler from coarse fraction rubber granules (%), and different lengths of resonator and filler holders.

Table 1 shows a list of the structure elements used and their geometric properties:

For this research, plates with different perforation diameter holes were chosen as the single-layer plates, with either C—central—or S—staged—perforations (Figure 5). The thickness of the plates is 10 mm, 15 mm, and 20 mm with perforations of 1.0 mm, 1.5 mm, and 2.0 mm. The resonators are of "X" and "O" shapes and are 50 mm or 100 mm in length. Resonators of different lengths and shapes were designed to manipulate sound waves at low frequencies. Different amounts (%) of sound-absorbing rubber filler and the position of the filler in the structure are also studied.

**Table 1.** Categories of sample parts and their geometrical parameters.

| Part | Type | Length/Thickness, mm | Perforation, mm | Perforation, % | Filler, % |
|---|---|---|---|---|---|
| Plate | Central | 10 | 1.0 | 7.4 | – |
| | Central | 15 | 1.0 | 7.4 | – |
| | Central | 20 | 1.0 | 7.4 | – |
| | Central | 10 | 1.5 | 12.0 | – |
| | Central | 15 | 1.5 | 12.0 | – |
| | Central | 20 | 1.5 | 12.0 | – |
| | Central | 10 | 2.0 | 13.6 | – |
| | Central | 15 | 2.0 | 13.6 | – |
| | Central | 20 | 2.0 | 13.6 | – |
| | Staged | 10 | 1.0 | 3.8 | – |
| | Staged | 15 | 1.0 | 3.8 | – |
| | Staged | 20 | 1.0 | 3.8 | – |
| | Staged | 10 | 1.5 | 4.8 | – |
| | Staged | 15 | 1.5 | 4.8 | – |
| | Staged | 20 | 1.5 | 4.8 | – |
| | Staged | 10 | 2.0 | 6.4 | – |
| | Staged | 15 | 2.0 | 6.4 | – |
| | Staged | 20 | 2.0 | 6.4 | – |
| Resonator | "X"-shaped | 50 | – | – | – |
| | "X"-shaped | 100 | – | – | – |
| | "O"-shaped | 50 | – | – | – |
| | "O"-shaped | 100 | – | – | – |
| Holder | – | 50 | – | – | – |
| | – | 100 | – | – | – |
| Filler | Rubber in different positions | – | – | – | From 9 to 50 |

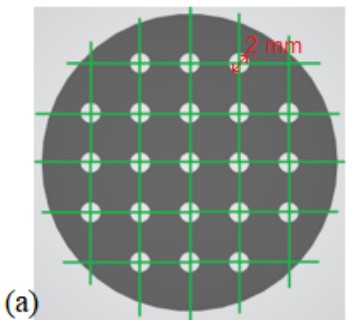 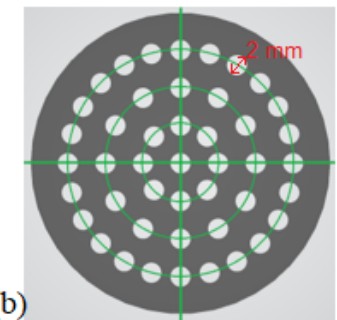

**Figure 5.** Staged (**a**) and central (**b**) 2 mm perforation of single layer plates.

An example of sample code used in this research to define the samples is presented in Figure 6.

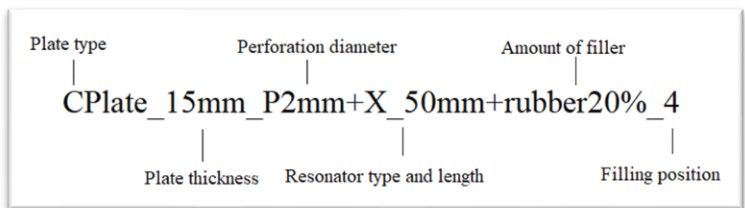

**Figure 6.** Coding example of the combined sample.

Figure 7 presents the results of sound absorption at different frequencies. Plates of 10 mm, 15 mm, and 20 mm thickness with perforations and their arrangement were measured. As we can see from the graph in Figure 7, the sound absorption peak of the 10 mm thick plate was obtained at 4000 Hz and is equal to 0.73. The best sound absorption was achieved with staged perforation with hole diameters of 1.5 mm when the total surface perforation is 4.8%. The worst absorption, at practically all frequencies, was obtained by applying a central perforation with hole diameters of 2.0 mm when a total surface perforation is 13.6%. In an overall comparison of perforations by diameter, the absorption peaks at 4000 Hz were obtained with the 1.5 mm perforation, with a sound absorption coefficient of 0.73 for the staged perforation and 0.66 for the central perforation. The worst results were obtained with 1.0 mm surface perforation, regardless of the placement. It is observed that staged perforation with different hole diameters showed better results, at practically all frequencies. However, it should be emphasized that, generally, poor absorption coefficients are achieved, and the use of only 10 mm thick plates against a rigid wall is not an advisable approach to reducing reverberation time in the room.

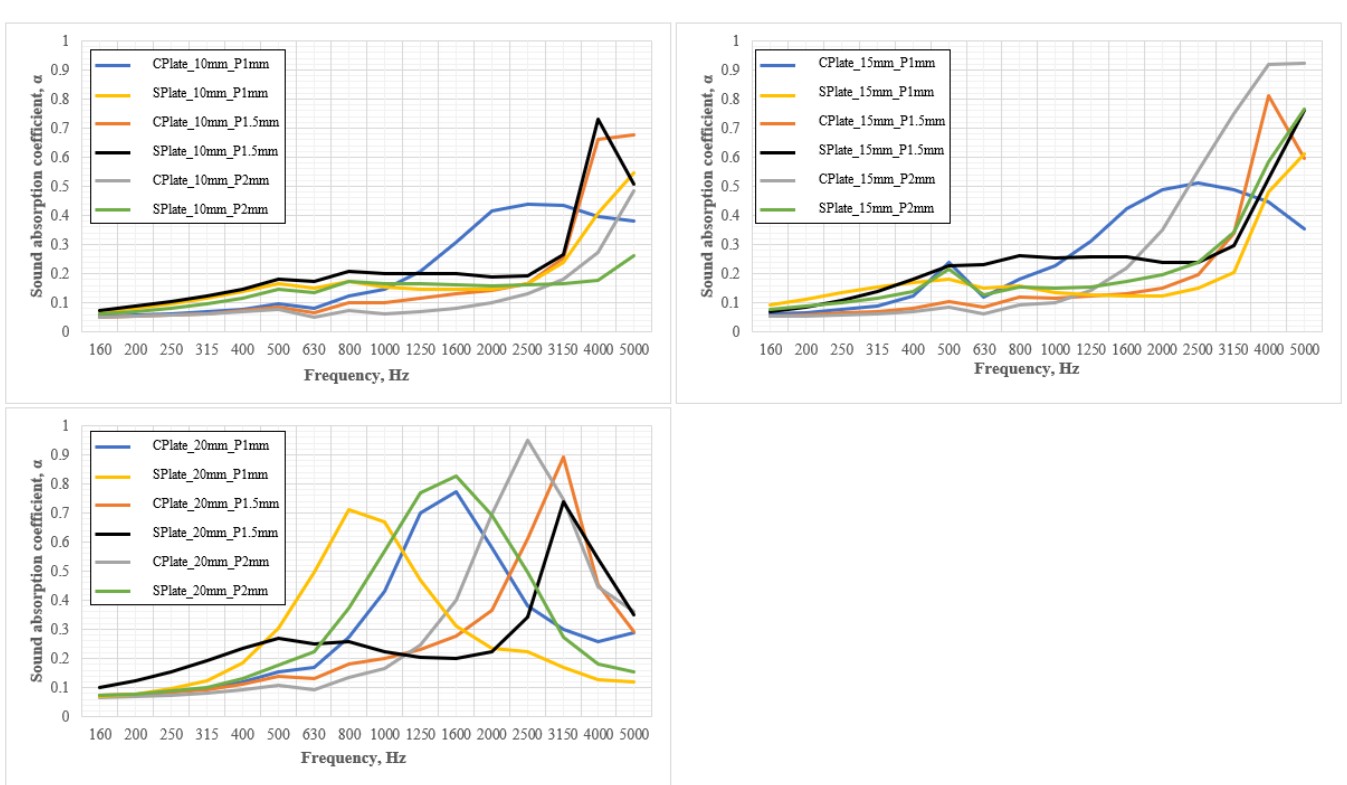

**Figure 7.** The sound absorption coefficients of 10 mm, 15 mm, and 20 mm thickness plates with 1.0 mm, 1.5 mm, and 2.0 mm diameter perforation at different frequencies.

The sound absorption peak of the 15 mm thick plate was obtained at 4000 Hz and is equal to 0.91. The best absorption was achieved with central perforation with hole diameters of 2.0 mm when the total surface perforation is 13.6%. The worst absorption, at low frequencies, was obtained with a central perforation with hole diameters of 2.0 mm, with a total surface perforation equal to 13.6%, and at high frequencies with a staged perforation with hole diameters of 1.0 mm with a total surface perforation equal to 3.8%. In an overall comparison of perforations by diameter, absorption peaks at 4000 Hz were obtained with 2.0 mm and 1.5 mm center perforations, achieving absorption coefficients of 0.91 and 0.80, respectively. The worst result was obtained with 1.0 mm surface perforation, regardless of the placement. It is observed that the central perforation with different hole diameters showed better results. It is emphasized that generally poor absorption coefficients at low frequencies are achieved with all plates. However, it is possible to use a

15 mm thick plate with central 2.0 mm hole diameters perforation against a rigid wall to absorb sound waves of 4000–5000 Hz.

The sound absorption peaks of the 20 mm thick plate obtained at 2500 Hz and 3150 Hz are equal to 0.95 and 0.89, respectively. The best sound absorption was achieved with central perforation with hole diameters of 2.0 mm when the total surface perforation is 13.6%. However, with this plate, the worst absorption was obtained at low frequencies, and, at high frequencies, the staged perforation plate with 1.0 mm hole diameters performed the worst. The worst results were obtained with 1.0 mm surface perforation, regardless of placement. Staged perforation with different hole diameters was observed to show better results at high frequencies. It should be emphasized that overall poor absorption coefficients at low frequencies are achieved with all plates. However, it is possible to use a central 20 mm thick plate with 2.0 mm hole diameters at the rigid wall to absorb 2500 Hz and a 1.5 mm hole diameter perforation to absorb 3150 Hz sound waves.

The absorption coefficients of "X" and "O" shape resonators (Figure 8) and different operating principles at different frequencies are presented in Figure 9.

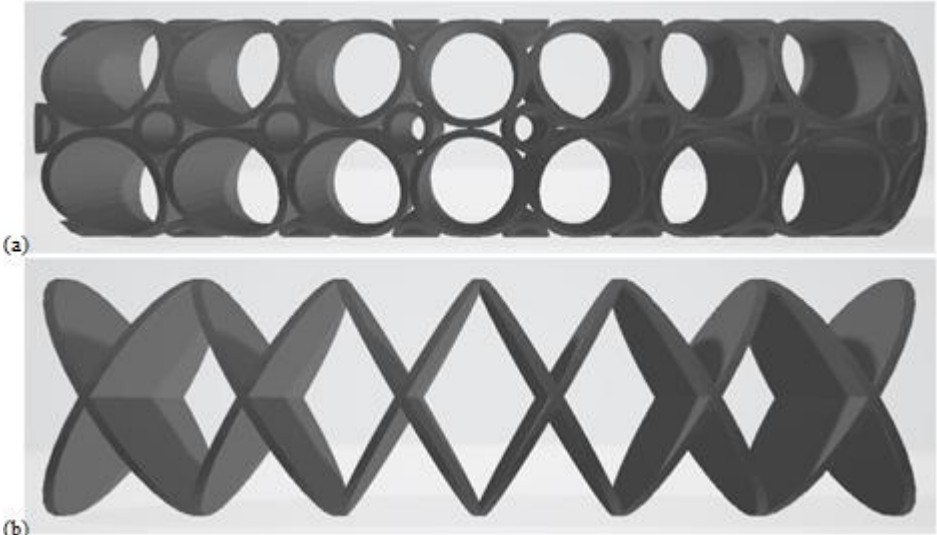

**Figure 8.** Side view of 100 mm long "O" (**a**) and "X" (**b**) shaped resonators.

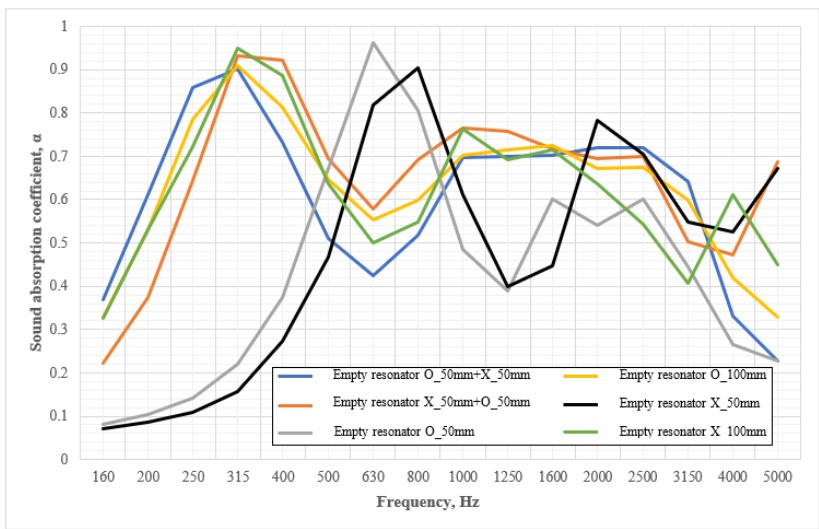

**Figure 9.** The sound absorption coefficients of 50 mm and 100 mm long empty "O" and "X"-shaped resonators at different frequencies.

The "X"-shaped resonator has spring-like properties, so the pressure of the sound wave is expected to excite it and absorb the lower frequency waves due to vibrations of the entire structure. The "O"-shaped resonator has circular air voids in its structure, which can act as Helmholtz resonators due to the pressure of the sound wave. The friction of the air particles in the resonator cavities is expected to absorb the sound waves.

As can be seen in Figure 9, different absorption peaks are obtained by utilizing resonators with different geometries. There are six structures measured, four 100 mm long and two 50 mm long. We can see the dependence of the absorption distribution on the length: the longer the structure, the more the absorption peak shifts to low frequencies. This happens because lower-frequency sound waves are longer, so long structures allow the low-frequency waves to interfere. Depending on the design and length of the selected resonator, constructive, partially destructive, or completely destructive interference of sound waves takes place. Because of this, the absorption peaks and dips at various frequencies are visible.

Both 50 mm long resonators showed similar results, but the "X" resonator has two peaks, 0.78 at high (2000 Hz) frequency and 0.90 at mid (800 Hz) frequency. The "O"-shaped resonator has a single peak at 630 Hz of 0.96, the highest sound absorption peak of any measured resonator. Equal sound absorption falls of up to 0.40 at 1250 Hz can be seen for both resonators of the "O" and "X" shapes.

The other two, 100 mm long "O" and "X"-shaped resonators, showed practically identical results, with the absorption coefficient peak at 315 Hz for the X resonator being slightly higher—0.94. It should be noted that using such a resonator on walls and building partitions is impractical due to the thickness of its construction, but using it on the ceiling of high rooms could be a suitable solution.

The symmetry of the combined resonator with both "X" and "O" shapes, with a total length of 100 mm (50 mm + 50 mm) was also measured, and it was observed that, if the "X" resonator is the first in the direction of sound wave travel, practically in all frequencies, better sound absorption results. From 800 Hz to 2500 Hz, the absorption coefficient remains above and around 0.70.

As we can see in Figure 10, the similarity of the graph was determined by the shapes and lengths of the resonators and their sound absorption. However, the worst results were obtained with a 10 mm thick plate with a staged 1.0 mm perforation. The diameter of the perforation holes in such a plate is too small, and the ratio of the perforation to the surface area is also low. Therefore, the sound wave cannot fully enter the structure and excite the resonator, which can further interfere with the sound waves and absorb them.

With 50 mm resonators, the best results, even with two sound absorption peaks, were obtained with a 10 mm thick plate with a central perforation of 2.0 mm. Combined with an "O"-shaped resonator, the first absorption peak is reached at the medium frequency of 630 Hz–0.92, and the high frequency of 3150 Hz–0.86. Combined with the resonator of "X" shape, the best structural combination cannot be singled out unequivocally, but there is an obvious difference in the sound absorption peak at 3150 Hz–0.87 of a structure with 10 mm thickness and 2.0 mm perforation with central arrangement plate. And the peak of the sound absorption of the lower frequency is shifted by one-third octave to a higher average frequency—630 Hz, and the sound absorption is 0.94. Staged perforation, in all cases, showed inferior results. The diameter of the perforation holes of such a plate and the surface area ratio are sufficient for the resonator to interfere with sound waves. As we can see, this happened at lower (from 500 to 630 Hz) and higher (from 1600 to 4000 Hz) frequencies. However, the resonator cannot effectively absorb 1000 Hz due to its natural frequency and the thickness of the plates. Therefore, we see a sharp drop at 1000 Hz and beyond 4000 Hz. This determines the operation of the resonator and its interaction with sound waves, but if the desired frequency range of sound absorption is close to 1000 Hz, applying such a design to a rigid support is not recommended.

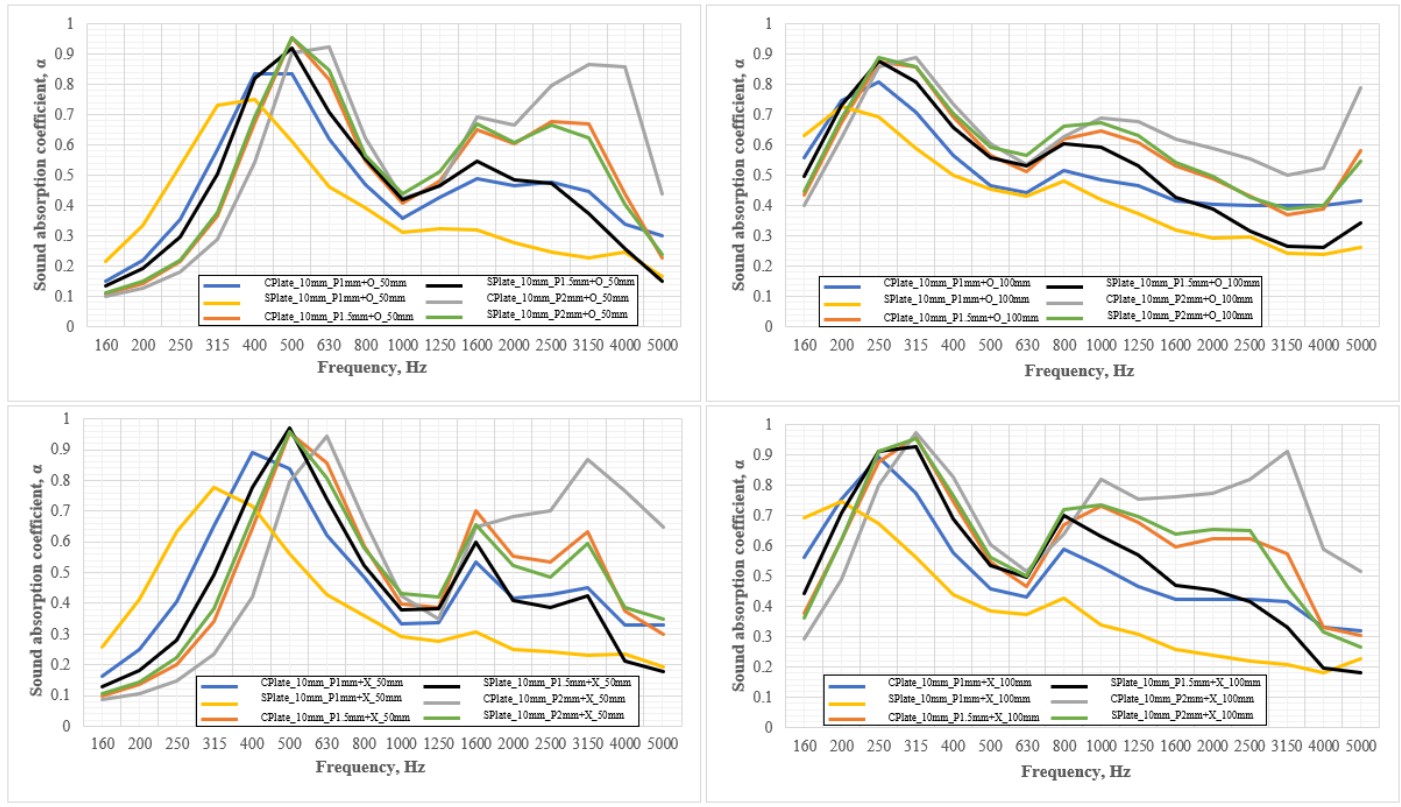

**Figure 10.** The sound absorption coefficients of 10 mm thick plates with different perforation arrangements (C—central; S—staged) combined with different resonators.

With 100 mm resonators, a 10 mm plate showed a slightly better sound absorption result in low frequencies, but an even drop is immediately visible when increasing the frequency of the sound wave. Combined with an "X"-shaped resonator, the best structural combination cannot be singled out unequivocally, but there is an obvious difference in the sound absorption peak at 3150 Hz–0.91 of a structure with 10 mm thickness and 2.0 mm perforation with central placement plate. This combination has two more peaks at 1000 Hz–0.82, and at 315 Hz–0.95. Despite the dip at 630 Hz, this design showed good overall results. As before, staged perforation, in all cases, showed inferior results. The resonator cannot effectively absorb 630 Hz due to its natural frequency and the thickness of the plates. Therefore, we see a sharp sound absorption drop at 630 Hz and beyond 3150 Hz. This determines the operation of the resonator and its interaction with sound waves, but, if the desired frequency range of sound absorption is close to 630 Hz, applying such a design to a rigid support is not recommended. Combined with an "O"-shaped resonator the best results were obtained with a 10 mm thick plate with a central perforation of 2.0 mm, with a sound absorption coefficient of 0.89 at 315 Hz. However, there is no big difference between all designs at lower frequencies, and the sound absorption efficiency was determined by the length of the resonator itself: the longer the resonator, the longer the sound wave it can absorb. As we can see from the graph, the smaller the perforation, the ratio of the surface area to the perforation, and the diameter of the holes, the worse the result. Therefore, the worst result is obtained at 1.0 mm perforation and is better at 2.0 mm perforation with central perforation placement.

As we can see from Figure 11, the similarity between the graphs was determined by the operation of the resonators and their sound absorption characteristics. Inferior results were obtained with a 15 mm thickness plate with a staged perforation of 1.0 mm at practically all frequencies. As before, staged perforation, in all cases, showed inferior results.

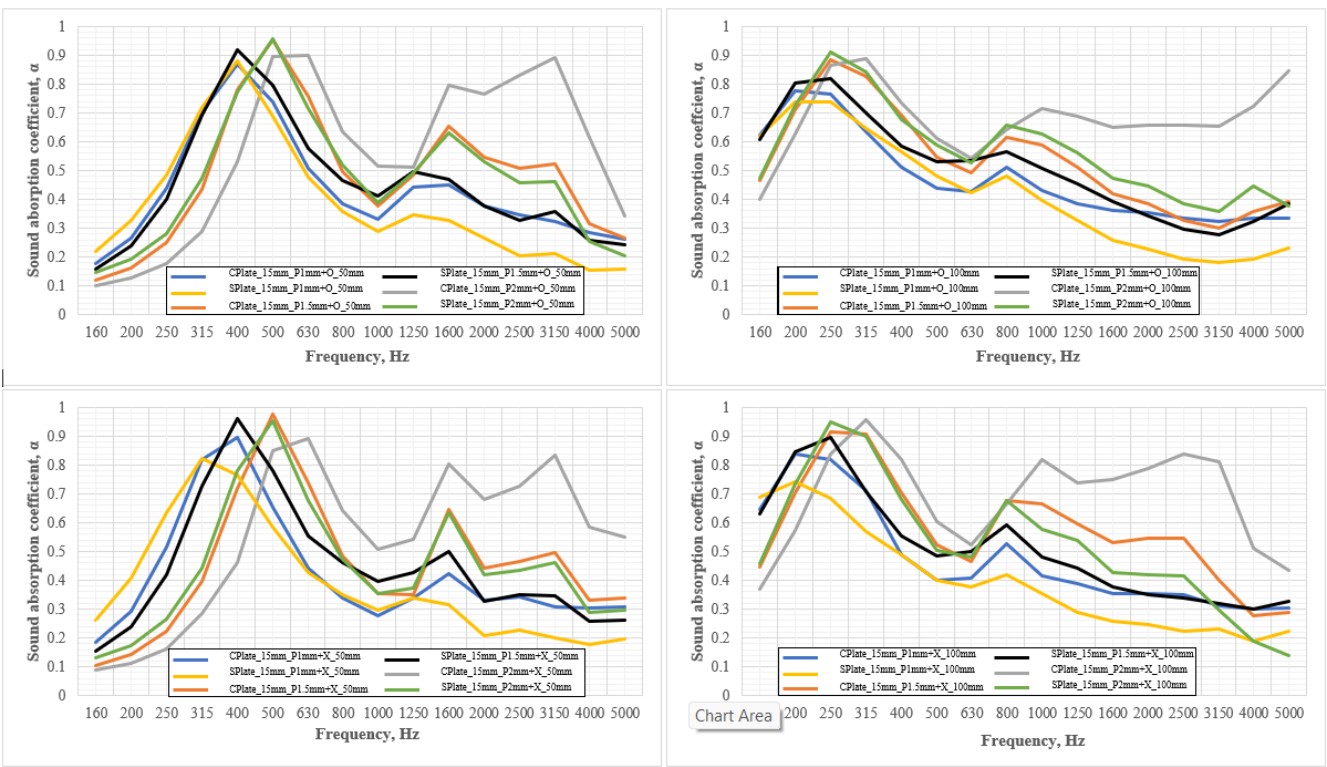

**Figure 11.** The sound absorption coefficients of 15 mm thick plates with different perforation arrangements (C—central; S—staged) combined with different resonators.

With 50 mm resonators, the best results, even with two sound absorption peaks, were obtained with a 15 mm thick plate with a central perforation of 2.0 mm. Combined with an "O"-shaped resonator the best structural combination cannot be singled out unequivocally, but there is an obvious difference in the sound absorption peak at 3150 Hz–0.89 of a structure with a 15 mm thick plate with a central perforation of 2.0 mm. Its low-frequency peak is also shifted by one-third octave to a higher average frequency of 630 Hz, with a sound absorption of 0.90. The highest peak among all samples of this type was obtained with a plate of 15 mm thickness with central perforation of 1.5 mm—at 500 Hz, the sound absorption coefficient is equal to 0.95. Combined with an "X"-shaped resonator due to the different perforation and arrangement of the plates, the sound absorption peaks are different at low frequencies. The highest absorption peak of 0.97 at 500 Hz was achieved with the design using 15 mm thick plates with 1.5 mm central perforation and 2.0 mm staged perforation. The sound absorption of these two structures is similar in all measured frequency ranges. At higher frequencies (from 1600 Hz), 15 mm thick plates with 2.0 mm central perforation showed significantly better sound absorption. Two sound absorption peaks are visible: at 1600 Hz–0.80 and 3150 Hz–0.83. The sound absorption efficiency of combined designs with 50 mm long resonators is largely dependent on the size of the plate perforation, which means that the worst result is obtained with a smaller diameter hole perforation, regardless of its placement. The resonator cannot effectively absorb 1000 Hz and 1250 Hz due to its natural frequency and the thickness of the plates. Therefore, we see a sharp sound absorption drop at 1000 Hz and beyond 3150 Hz. This determines the operation of the resonator and its interaction with sound waves, but, if the desired frequency range of sound absorption is close to 1000 Hz, applying such a design to a rigid support is not recommended.

With 100 mm resonators at high frequencies, the difference in the sound absorption coefficient was determined by the arrangement and size of the plate perforations. The larger the size of the perforation holes, the better the higher frequencies sound absorption peak or average sound absorption. With an "O"-shaped resonator, comparing 1.0 mm

and 2.0 mm central perforation, it was found that, from 1600 Hz to 3150 Hz, the sound absorption coefficient remains at 0.65 with the 2.0 mm perforation and around 0.35 with the 1.0 mm perforation. All samples peaked at 250 Hz, when the highest peak was obtained with a 15 mm thick plate with a staged 2.0 mm perforation, which equals 0.91. The peak shift at a low frequency of 250 Hz for all samples was determined by the length of the resonator, which allows lower frequency (longer sound wave) sound waves to be absorbed more efficiently. Comparing 1.5 mm and 2.0 mm central perforations, combined with an "X"-shaped resonator, it was found that from 1000 Hz to 3150 Hz with a 2.0 mm perforation, the sound absorption coefficient is about 0.80, while with a 1.5 mm perforation, it decreases from 0.66 to 0.55. As before, staged perforation, in all cases, showed inferior results. For all samples, the peak is seen at low frequencies, when the highest peak was obtained with a plate having a thickness of 15 mm and a central perforation of 2.0 mm and equal to 0.95. At high frequencies, the difference in the sound absorption coefficients was determined by the arrangement and size of the plate perforations.

As we can see in Figure 12, the similarity between the graphs was determined by the operation of the resonators and their sound absorption characteristics. However, due to the different perforations and arrangements of the plates, the sound absorption peaks are different at low frequencies.

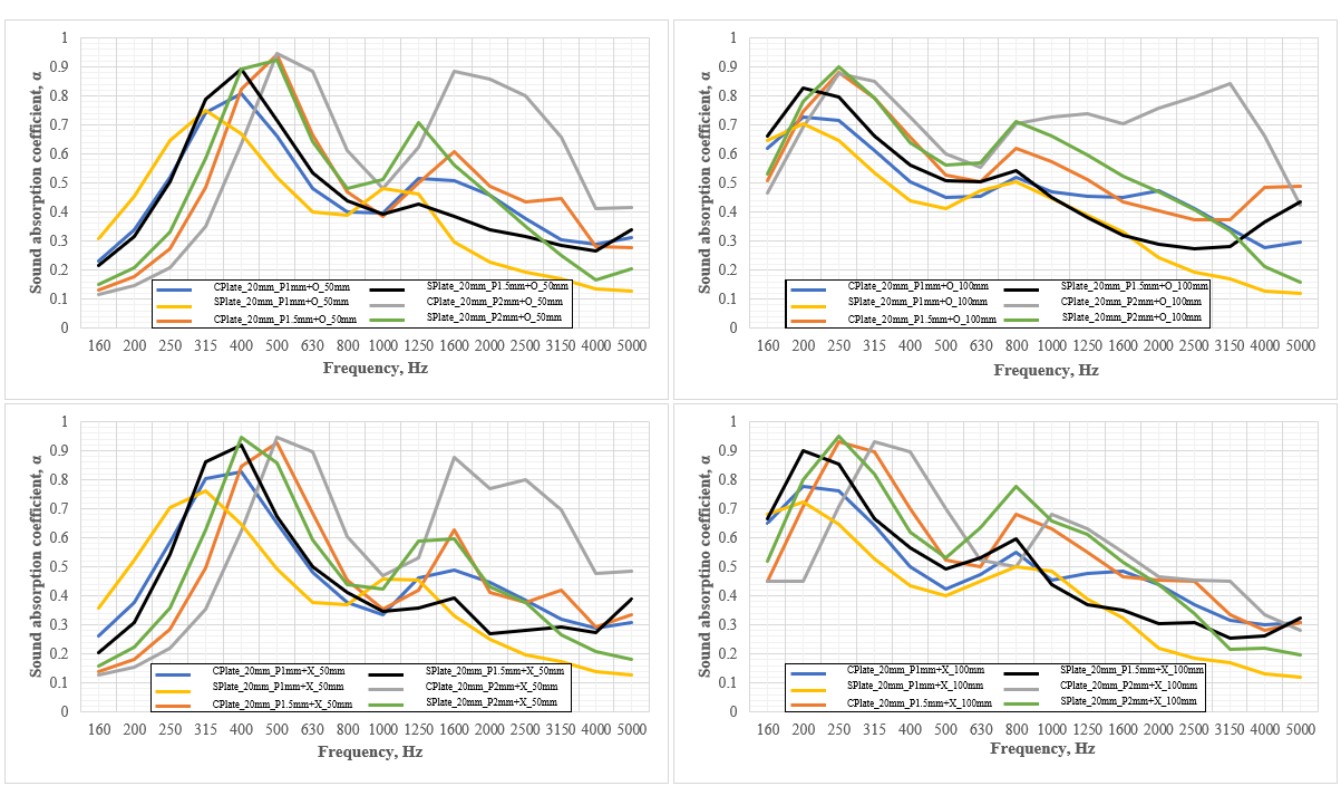

**Figure 12.** The sound absorption coefficients of 20 mm thick plates with different perforation arrangements (C—central; S—staged) combined with different resonators.

With 50 mm resonators, the worst results were obtained with a 1.0 mm staged perforation at almost all measured frequencies. The sound absorption efficiency of this combined design is largely dependent on the size of the perforation, which means that the worst result is obtained with a smaller diameter hole perforation, regardless of its placement. With an "O"-shaped resonator, the highest absorption peak of 0.94 is achieved at 500 Hz with the design using a 20 mm thick plate with 1.5 mm central perforation, 2.0 mm staged perforation and 2.0 mm central perforation. At higher frequencies (from 1600 Hz), a 20 mm thick plate with a central perforation of 2.0 mm showed significantly better sound absorption. The visible sound absorption peak at 1600 Hz is 0.90. With the "X"-shaped resonator,

the two highest absorption peaks are achieved with the design using a 20 mm thick plate with 2.0 mm central perforation—0.95 at 500 Hz and 0.87 at 1600 Hz. At higher frequencies (from 1600 Hz), the 20 mm thick plate with 1.5 mm and 2.0 mm central perforation and 2.0 mm staged perforation showed significantly better sound absorption. This was due to the percentage of perforations of the surface layer of the plates and the size of the holes, which allowed high-frequency waves to interact with the resonator inside the structure.

With 100 mm resonators, the shift of the peak toward lower frequencies for all samples was determined by the length of the resonator, which allows lower frequencies (longer wavelength) waves to be efficiently absorbed. At high frequencies, the difference in the sound absorption coefficients was determined by the arrangement and size of the plate perforations. The larger the size of the perforation holes, the better the high-frequency sound absorption peak or overall absorption average. With the "O"-shaped resonator, comparing 2.0 mm central and staged perforations, it was found that from 800 Hz to 3150 Hz with central perforations, the sound absorption coefficient varies from 0.70 to 0.84, and with staged perforations, it decreases steadily from 0.70 to 0.30. With the "X"-shaped resonator, when comparing 1.5 mm and 2.0 mm central perforation, it was found that the sound absorption coefficient at 800 Hz varied from 0.68 to 0.77. As before, staged perforation, in all cases, showed inferior results. For all measured structures, the second peak appears after a sharp drop in sound absorption at 500 Hz and 630 Hz, due to the natural frequency of the samples. However, the larger the diameter of the perforation holes and the ratio of the perforation to the surface, the more clearly visible the drop in sound absorption and the sharp rise after it. In addition, a higher perforation ratio in this case means a higher high-frequency sound absorption peak.

Figures with graphs of the results and pictures with different amounts (%) and positions of the rubber granules in the resonator structure are presented below. Coarse fraction rubber granules (Figure 13) were used for the sound-absorbing filler, which, depending on the design and position in the resonator, showed different results in absorbing sound waves at certain frequencies.

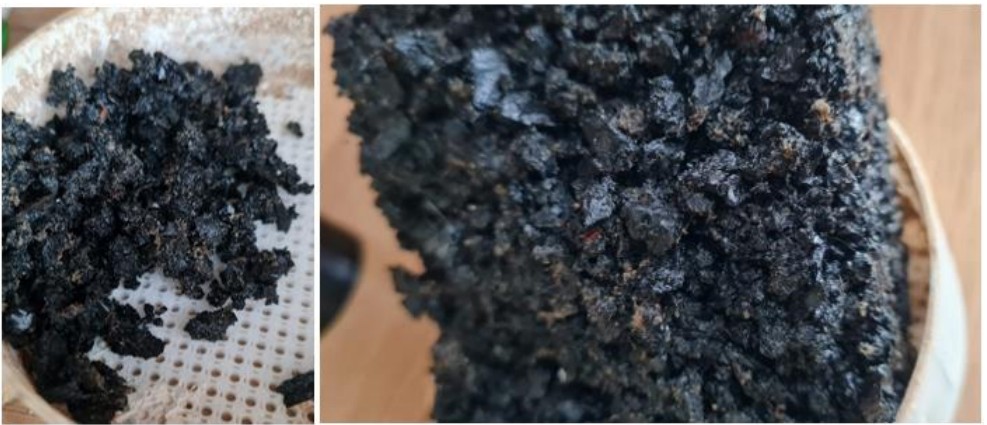

**Figure 13.** Coarse fraction rubber granules used as a sound-absorbing filler.

Figure 14 shows the results of a 50 mm long "O"-shaped resonator with 16%, 33%, and 50% rubber filler of the resonator volume. Three different filler positions were studied using 16% rubber filler in a 50 mm long "O"-shaped resonator. Up to a frequency of 630 Hz, there is practically no difference in the adsorption coefficient between the positions. This means that the absorption of the sample filler is low. We can see this when we compare the sample with an empty resonator of the same type. The absorption peak of all three samples, regardless of the filler position, was obtained at 630 Hz and reaches 0.95. However, a clear difference is seen at high frequencies, where the best result is achieved with the third filler position, where the second absorption peak reaches 0.82 at 2500 Hz. The worst result was achieved with the first position of the filler, when the filler is closest to the sound

source, which prevents the "O"-shaped resonator from fully functioning and absorbing sound waves.

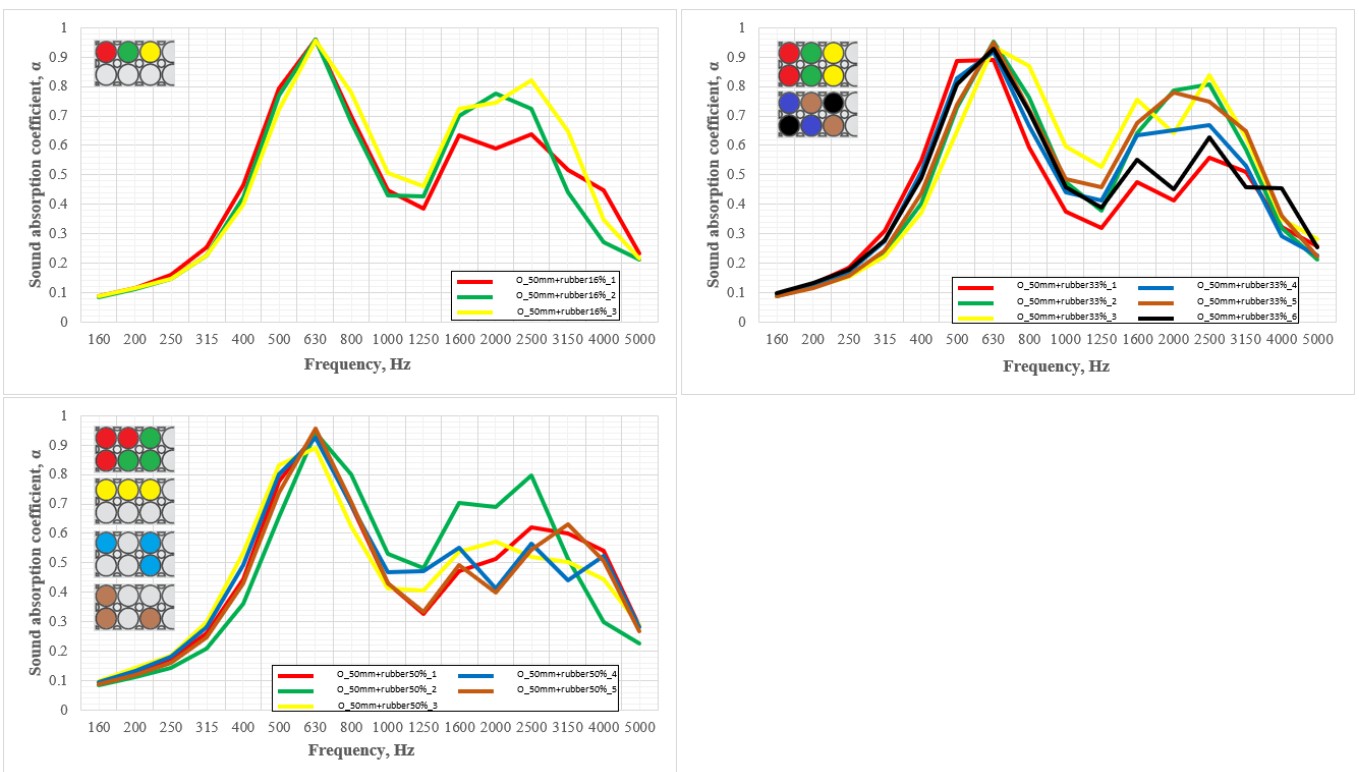

**Figure 14.** The sound absorption coefficients of a 50 mm long "O"-shaped resonator with different rubber granules amounts and positions at different frequencies.

Six different filler positions were studied using 33% rubber filler in a 50 mm long "O"-shaped resonator. Up to a frequency of 630 Hz, there is practically no difference in the adsorption coefficient between the positions. This means that the absorption of the sample filler is again low. We can see this when we compare the sample with an empty resonator of the same type. The absorption peak of all six samples, regardless of the filler position, was obtained at 630 Hz and varies between 0.88 and 0.93. However, immediately after the absorption peak, a clear difference is seen at higher frequencies. In this case, structures with fillers in the second, third, and fifth positions were superior, where the best result is achieved with the third filler position, where the second absorption peak reaches 0.83 at 2500 Hz. Of the structures with the first, fourth, and sixth positions of the filler, the worst result was with the first position, when the filler is closest to the sound source, which does not allow the "O"-shaped resonator to fully function and absorb sound waves. Although the high-frequency sound absorption is different, the same drop in sound absorption at 1250 Hz is seen for all designs, with the worst cases being down to 0.31 in the first filler position.

Five filler positions were studied using 50% rubber filler in a 50 mm long "O"-shaped resonator. Once again, up to a frequency of 630 Hz, there is practically no difference in the result. The absorption peak of all five samples, regardless of the filler position, was obtained at 630 Hz and varies between 0.88 and 0.95. However, immediately after the peak, the difference is visible at higher frequencies, especially from 1000 Hz. The design with the second filler position proved to be the best, in which the second sound absorption peak appeared at 2500 Hz and reaches 0.80. As before, the best absorption is achieved when the filler position is farthest from the sound source, which allows the resonator to interfere with the waves. Although the high-frequency sound absorption is different, the same drop

in sound absorption at 1250 Hz is seen for all designs, with the worst cases being down to 0.33 for the first and fifth filler positions.

Figure 15 shows the results of a 100 mm long "O"-shaped resonator with 14%, 28%, and 50% rubber filler of the resonator volume. Three filler positions were studied using 14% rubber filler in a 100 mm long "O"-shaped resonator. Up to a frequency of 315 Hz, there is practically no difference in the result. The absorption peak of all three samples, regardless of the filler position, was obtained at 315 Hz and reaches 0.90. However, the difference is visible at high frequencies, where similar results are obtained with the second and third filler positions, where the second absorption peak reaches 0.89 at 3150 Hz. The difference may not be visible as a result of the very small amount of rubber filler in the structure. The worst result was achieved with the first position of the filler, in which the filler is closest to the sound source, which prevents the "O" resonator from fully functioning and absorbing sound waves. The entire structure of the resonator behind the filler does not optimally absorb sound waves.

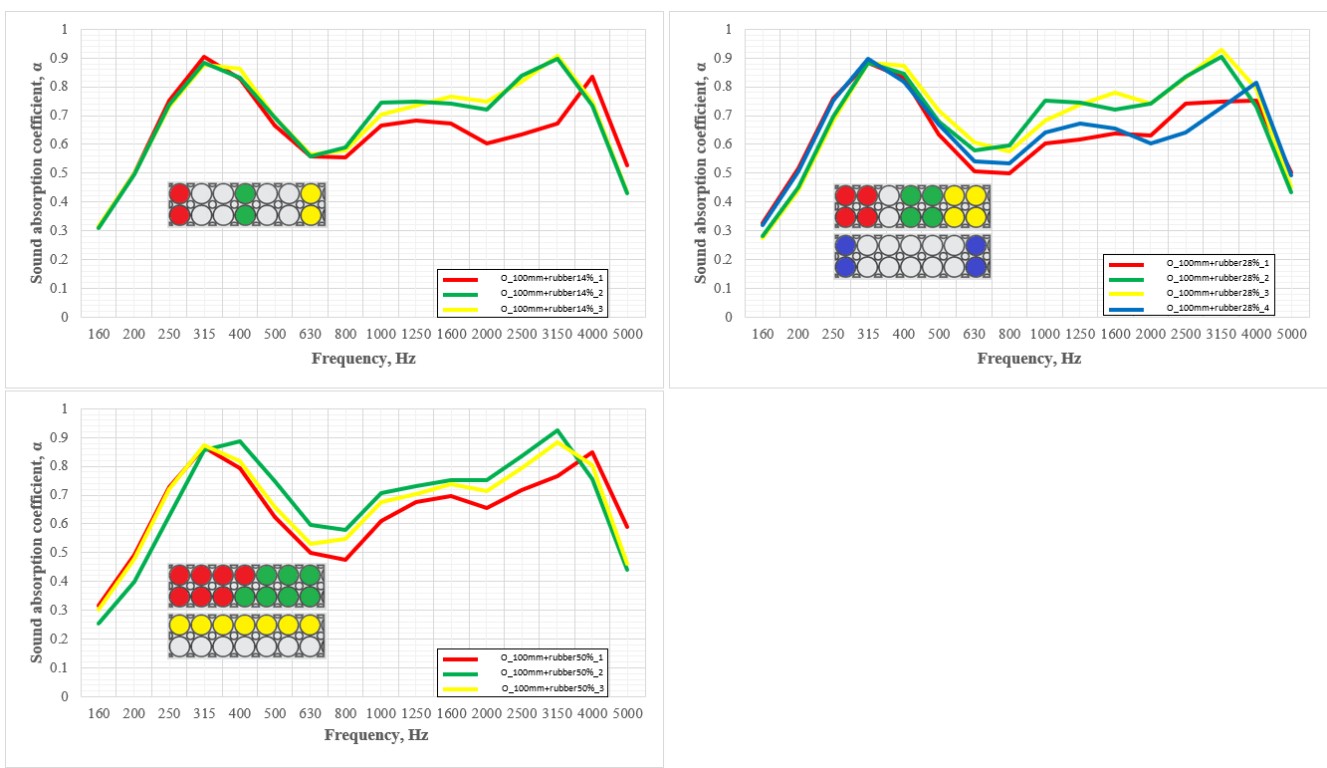

**Figure 15.** The sound absorption coefficients of a 100 mm long "O"-shaped resonator with different rubber granules amounts and positions at different frequencies.

Four filler positions were studied using 28% rubber filler in a 100 mm long "O"-shaped resonator. Up to 315 Hz, the result is practically the same. The absorption peak of all four samples, regardless of the filler position, was obtained at 315 Hz and reaches 0.89. However, a noticeable difference is seen at high frequencies, where similar results are obtained with the second and third filler positions, where the second absorption peak reaches 0.93 at 3150 Hz. The difference may not be visible due to the small amount of rubber filler in the structure. The worst result was achieved with the first and fourth positions of the filler, in which the filler is closest to the sound source, which prevents the "O"-shaped resonator from fully functioning and absorbing sound waves. Then, the entire resonator structure behind the filler does not optimally absorb sound waves, which reduces sound absorption at high frequencies.

Three filler positions were studied using 50% rubber filler in a 100 mm long "O"-shaped resonator. Up to a frequency of 315 Hz, there is practically no difference in the

result. We can also see this when comparing samples with the same empty resonator, but its peak is slightly higher (0.90 at 315 Hz), but the efficiency at high frequencies is worse. The absorption peak of all three samples, regardless of the filler position, was obtained at 315 Hz and reaches 0.87. However, the difference is visible at high frequencies, where the best result is achieved with the third filler position, where the second absorption peak reaches 0.92 at 3150 Hz. The difference is visible due to the sufficient amount of rubber in the structure and its position in the resonator, and compared to the empty sample, it can be said that the rubber granules in the structure helped to absorb the high-frequency sound waves. The worst result was achieved with the first position of the filler, in which the filler is closest to the sound source, which does not allow the "O"-shaped resonator to function optimally and absorb sound waves. However, the high-frequency sound absorption peak is shifted to a higher frequency of 4000 Hz and is equal to 0.85.

Figure 16 shows the results of a 50 mm long "X"-shaped resonator with 10%, 20%, 30%, and 50% rubber filler of the resonator volume. Three filler positions were studied using 10% rubber filler in a 50 mm long "X"-shaped resonator. Due to its "X" design, the resonator allows even a small volume (lower mass) filler to act on the sound waves. As we can see from the graph, the structure with the first and second positions of the rubber filler reached its peak sound wave absorption of about 0.95 at 630 Hz, and the structure with the third position of the filler at 800 Hz reached its peak of 0.93. This was due to the distance of the position of the filler from the sound source in the structure itself. This means that having the sound-absorbing material closer to the sound source in the structure shifts the peak towards lower frequencies, while, having it further away, the sound absorption is determined by the design and operation of the resonator. Therefore, in the case of the first position, we see poor absorption results in the entire frequency range beyond the peak. While, when comparing the second and third positions, we can see that there is no sudden drop in absorption in the third position, due to the natural frequency of the structure, but the second absorption peak at high frequencies is more clearly visible in the second position. Due to the placement of the filler, a resonant frequency is captured, depending on whether or not the sound absorption is enhanced or attenuated in the material. For the third position, there was no sharp drop at 1250 Hz, so there is no visible sharp rise, and there is no advantage in sound absorption at high frequencies due to the resonant frequency effect. Although the structures with the filler in the first position showed a slightly better sound absorption coefficient at low frequencies, the overall average sound absorption is significantly worse. The second position, due to its two sound absorption peaks, can be considered better; but in this case, there is no unambiguously best structure.

Five filler positions were studied using 20% rubber filler in a 50 mm long "X"-shaped resonator. As we can see from the graph, the position of the sound-absorbing layer in the resonator structure influences the sound absorption peak at low frequencies. Having the sound-absorbing material closer to the sound source in the structure shifts the peak towards lower frequencies, but at higher frequencies, the sound absorption is determined by the design of the resonator and its operation. Therefore, in the case of the first position, we see poor absorption results in the entire frequency range beyond the peak. Due to the two peaks of sound absorption, the best results were obtained by applying the sound-absorbing material in the fourth position in the structure. The first peak of sound absorption occurs at an average frequency of 800 Hz and is equal to 0.95, while the second peak occurs at 2000 Hz and is equal to 0.86. A repeated fall and rise in the sound absorption coefficient is clearly visible, which determines the appearance of the resonant frequency in the structure and the operation of the resonator itself. A 50 mm long "X"-shaped resonator with 20% rubber granule filler shows a sound absorption trend in which, the closer the filler is to the resonator, the higher the sound absorption peak at lower frequencies. However, this results in poorer results at high frequencies. Due to the design of the resonator, the action of the sound wave and the natural frequency of the structure lead to different sound absorption drops.

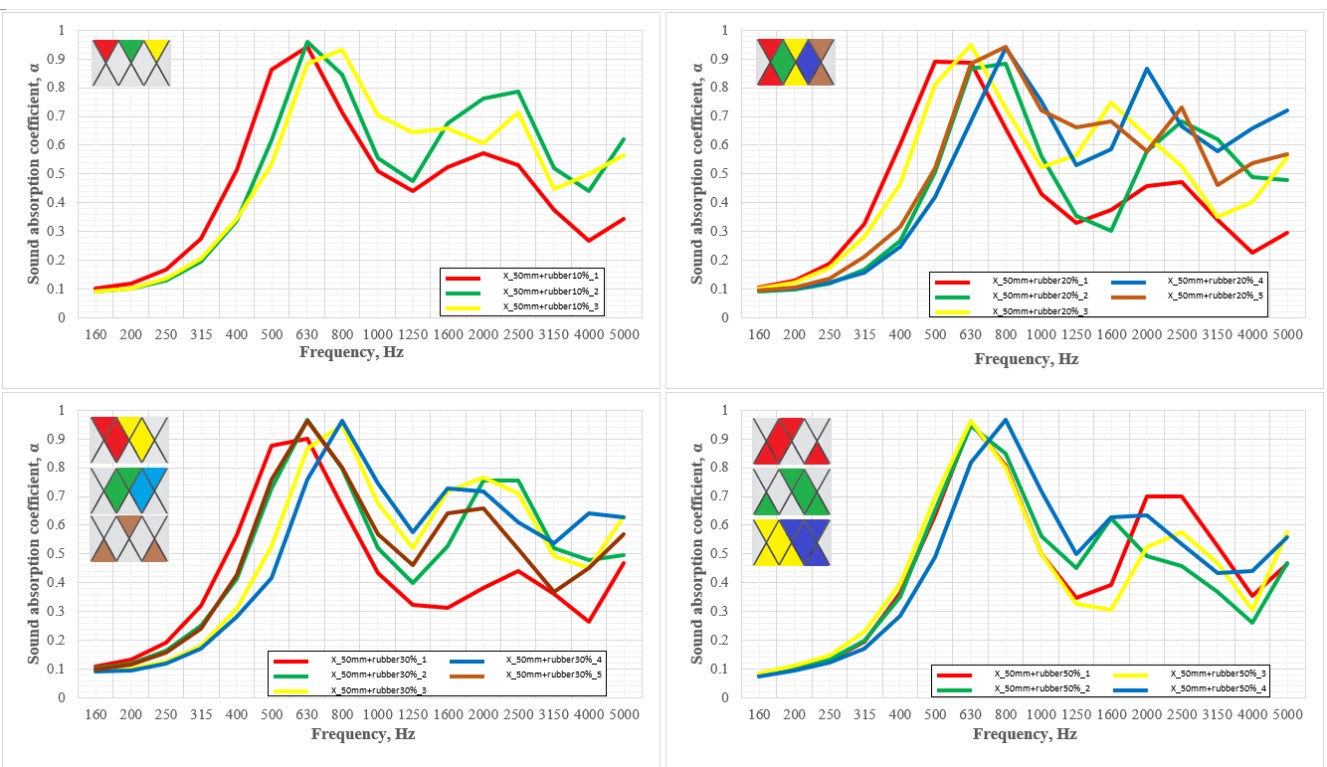

**Figure 16.** The sound absorption coefficients of a 50 mm long "X"-shaped resonator with different rubber granules amounts and positions at different frequencies.

Five filler positions were studied using 30% rubber filler in a 50 mm long "X"-shaped resonator. As we can see from the graph, having 20% or 30% filler content of the first position in the structure practically does not change the results. This is because the filler does not transmit the sound waves to the resonator but immediately interacts with the material, which then ineffectively absorbs the sound waves. The first position performs the worst but achieves a sound absorption peak of 0.89 at 630 Hz. There is no obvious best design in terms of sound absorption, but good sound absorption peaks at 630 Hz and 800 Hz are achieved. The second and fifth positions have a sound absorption peak at 630 Hz of about 0.96 and the third and fourth positions of about 0.96 at 800 Hz. All structures, except the structure with the filler in the first position, show a sharp drop in sound absorption due to the natural frequency of 1250 Hz.

Four filler positions were studied using 50% rubber filler in a 50 mm long "X"-shaped resonator. As we can see from the graph when applying a 50% volume filler in the resonator structure, there is practically no difference at low frequencies up to 630 Hz, except for with the filler in the fourth position, because being further from the sound source, the filler allows the resonator to work with sound waves, but then the sound absorption peak is shifted to 800 Hz. A sharp drop in sound absorption due to the natural frequency of 1250 Hz is visible for all structures. Here, the best design cannot be selected, which indicates that it is not appropriate to further increase the amount (volume) of the filler in the material.

Figure 17 shows the results of a 100 mm long "X"-shaped resonator with 9%, 27%, and 45% rubber filler. Six filler positions were studied using 9% rubber filler in a 100 mm long "X"-shaped resonator. As we can see from the graph, the trends were determined by the "X"-shaped 100 mm long resonator, and the peaks and troughs of sound absorption were indicated by the positions of the rubber beads in the resonator structure. The worst result was the application of the rubber granule layer in the fourth position, with the lowest sound absorption peak at 250 Hz of 0.86, and the worst results at the other measured frequencies. As before, this is because the sound-absorbing layer is closer to the sound source than the empty cavities of the resonator. In this way, sound waves are not optimally absorbed,

which makes the resonator itself work inefficiently. Here, the best result cannot be singled out unequivocally, due to the similarity in the absorption coefficients across all frequencies. This may mean that even when deeper in the structure from the sound source, the total mass of the sound wave absorbing layers is too small to affect the sound absorption, which we can see by comparing the sound absorption with an empty resonator of the same design. However, the highest sound absorption peak obtained was around 0.93 at 315 Hz for all other 9% rubber filler positions.

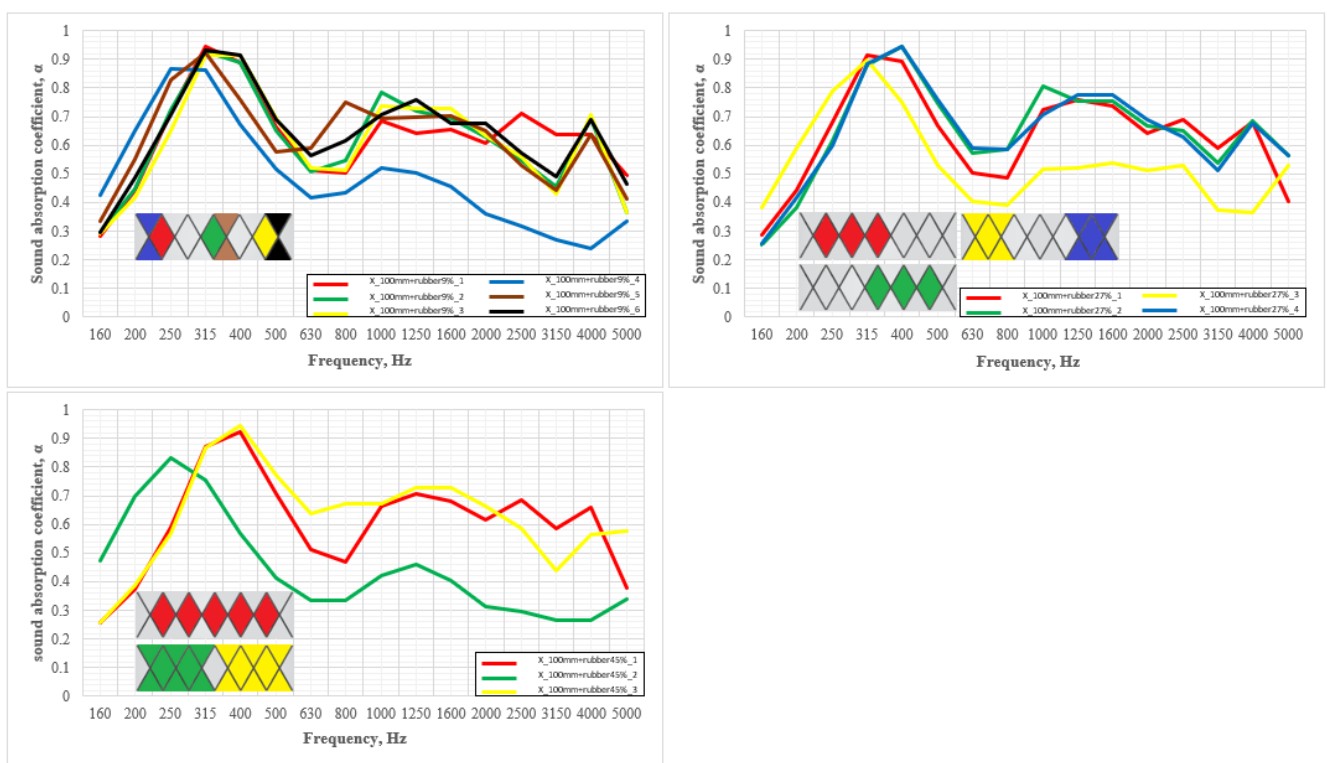

**Figure 17.** The sound absorption coefficients of a 100 mm long "X"-shaped resonator with different rubber granules amounts and positions at different frequencies.

Four filler positions were studied using 27% rubber filler in a 100 mm long "X"-shaped resonator. As we can see from the graph, the results of the third position are the worst, when the sound absorption coefficient is equal to 0.90 at 315 Hz. This position is closest to the sound source, which prevents the resonator from working efficiently and absorbing sound waves of different frequencies. Next worst was having the absorbent layer in the first position. The graph curve is similar to that of the third position, but with slightly better results in the higher frequencies after the peak. The sound absorption peak occurred at 315 Hz and was equal to 0.92. The second and fourth position variants had better and similar results, in which the sound absorption peak was at 400 Hz and about 0.92.

Three filler positions were studied using 45% rubber filler in a 100 mm long "X"-shaped resonator. As we can see from the graph, the results of the sound absorption are similar for the second and third positions of the rubber granules in the resonator structure. This means that the amount of filler in the structure, at 45% of the volume of the resonator, has little effect on sound absorption; however, the position of the filler must be such that it allows sound waves to interact with the resonator itself. This is why we see such poor results with the first rubber filler position, which is closest to the sound source, preventing the "X"-shaped resonator from fully functioning and absorbing sound waves. However, a sound absorption coefficient of 0.83 at 250 Hz was achieved. When using the second and third positions, the sound absorption peak was reached at a frequency of 400 Hz and equal to about 0.94. At higher frequencies, a drop is seen and all curves are similar, which

dictated the design and principle of operation of the resonator. For sound absorption and improvement of reverberation or speech intelligibility in a room, it is not advisable to use such a structure, unless the goal is to reduce the reverberation time of 400 Hz specifically, but the same result or slightly better can be achieved with smaller amounts of rubber granule filler.

Below, the best sound-absorbing structures and their measured results are selected and presented for comparison and general conclusions. In all cases, the best results were obtained using central perforation plates with a 2.0 mm diameter perforation, regardless of thickness. In the following graphs, the results were distributed according to the thickness of the plates used.

As we can see in the graph in Figure 18, when applying a 10 mm thick 2.0 mm central perforation plate, the curves were distributed along the length of the resonator. The longer the resonator, the better it performs at lower frequencies. This happens because of the length of the sound wave—the lower the frequency, the longer the sound wave. Therefore, we can see that the combined design with a 100 mm long resonator, regardless of whether it is "X"- or "O"-shaped, reached a good sound absorption peak at 315 Hz. The difference between the "X" and "O" shapes is only visible at high frequencies, above 1600 Hz, in which, with the "O" resonator, the sound absorption decreases steadily with increasing frequency of the sound waves. The second peak of the 100 mm "X" design resonators is reached at 3150 Hz, and differences in the curves are only visible due to the different amounts of rubber filler in the structure. Having 27% rubber filler in the second position helped achieve better sound absorption at 3150 Hz compared to having 9% in the second position: at 27%–0.94, and 9%–0.89. However, no significant differences were observed across the entire measured frequency range, so it may not be economically justified to use a higher amount of rubber filler to obtain better sound absorption. By comparing combined structures with the 50 mm long "X" and "O"-shaped resonators, different sound absorption peaks and high-frequency results were achieved. A higher sound absorption peak of 0.95 was achieved at 630 Hz with the "X" resonator and higher absorption at all high frequencies. The 33% rubber filler to the total volume of the resonator in the structure in the third position helped to raise the sound absorption peak and slightly increased the absorption throughout the all-frequency range. The sound absorption capacity of the "O"-shaped resonator is significantly worse compared to those of other combined selected resonators.

By using a 15 mm thick plate with 2.0 mm central perforation, the curves were distributed along the length of the resonator. Therefore, we can see that the combined design with a 100 mm long resonator, regardless of whether it is "X"- or "O"-shaped, reached a good sound absorption peak at 315 Hz. The difference between the "X" and "O" shapes is only visible at high frequencies, above 1600 Hz, in which, with the "O" resonator, the sound absorption decreases steadily with increasing frequency of the sound waves. Compared to the previous design, where a 10 mm thick plate was used and the sound peak was at 3150 Hz in the high frequencies, here with a 20 mm thick plate the peak is at 2500 Hz and equal to 0.84 to 0.88, depending on the amount of rubber. The shift in the frequency band is due to the overall thickness of the structure, which is why there was not much difference between the samples, depending on the amount of rubber, because of the slightly lower frequency. By comparing combined structures with the 50 mm long "X"- and "O"-shaped resonators, different sound absorption peaks and high-frequency results were achieved. The same sound absorption peak of 0.94 was achieved at 500 Hz with the "O"-shaped resonator and at 630 Hz with the "X"-shaped resonator. The "X"-shaped resonator has better absorption at all frequencies above 1600 Hz. Having 33% rubber filler in the structure in the third position helped to raise the sound absorption peak and slightly increased the absorption throughout the frequency range. The sound absorption capacity of the "O"-shaped resonator is worse compared to that of other combined selected resonators.

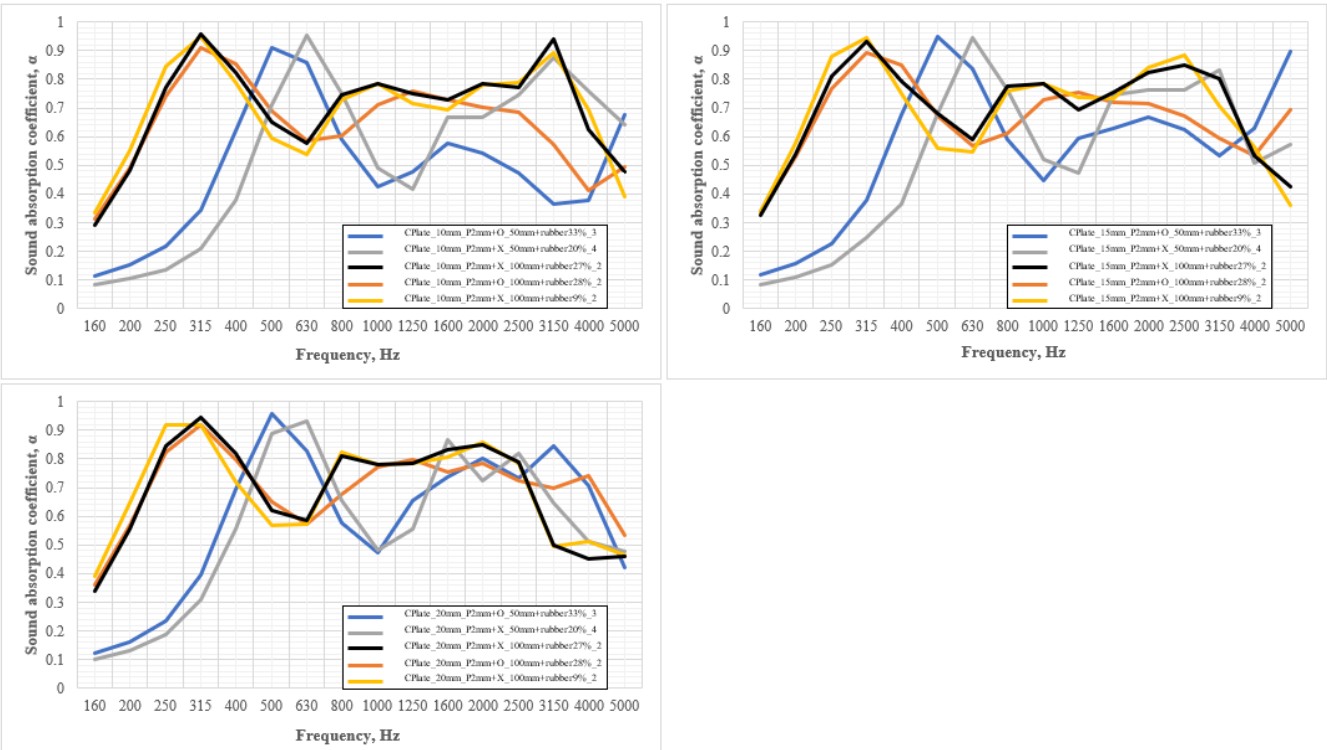

**Figure 18.** Sound absorption coefficients of plates 10 mm, 15 mm, and 20 mm thick with central perforation arrangement and different resonators with different rubber fillers in different positions.

Applying a 20 mm thick plate with 2.0 mm central perforation, the curves were distributed along the length of the resonator. Therefore, we can see that the combined design with a 100 mm long resonator, regardless of whether it is "X" or "O"-shaped, achieved a good sound absorption peak at 315 Hz of 0.94. Here, there were no significant differences because of the amount of filler in the high frequencies. Comparing the 50 mm long "X"- and "O"-shaped combined structure, different sound absorption peaks and high-frequency results were achieved. A slightly better sound absorption peak of 0.95 was achieved at 500 Hz with the "O"-shaped resonator and 0.93 at 630 Hz with the "X"-shaped resonator. The "X"-shaped resonator also achieved a second good sound absorption peak at 1600 Hz–0.86.

As noted earlier, a design with a longer resonator absorbs sound better at lower frequencies. The longer the structure, the more the absorption peak shifts to low frequencies. Such a structure is also less affected by the resonant frequency due to the interference and diffraction of sound waves inside the structure itself. The rubber granules in the resonator, depending on their position in the structure, can improve the absorption of high frequencies, or, if there are more of them, push the absorption of sound toward low frequencies. Although the 100 mm long resonator absorbs lower frequencies better, due to its length/thickness, it is not advisable to integrate it into building rooms, unless it is necessary to absorb a specific frequency, in which case it is recommended to use it on the ceiling.

## 4. Discussion

Integrating this product into the building and construction sector would extend the life cycle of plastic and rubber waste. Combined structures with plastic panels, plastic resonators, and rubber fillers would be a good alternative to existing solutions in building construction. As an example, the sound absorption coefficients of wall/ceiling materials such as standard gypsum plasterboard vary only from 0.30 to 0.10, depending on the frequency, while plywood has absorption coefficients of only 0.4 to 0.10, depending on the

frequency. With the correct geometric parameters, the structures proposed in the current work should provide better sound absorption than traditional solutions when used as wall panels.

It would be interesting to manufacture a real-scale wall panel from recycled plastic and rubber and investigate its acoustic parameters; then recycle its components, sort them, and manufacture the same panel again. The comparison between these two panels would show how well this concept would work and what recycling technologies are the best for this kind of wall panel. However, some researchers think that large-scale manual manufacturing potentially leads to a loss of consistency and unstable product quality because such designs mainly rely on manual manufacturing and 3D printing technologies. In addition, when it comes to large-scale production, 3D printing technology may suffer from high-cost consumption and low production efficiency [29].

In future research, it may be interesting to investigate the sound insulation properties of wall panels or structures with a similar working resonance principle. Due to their good sound absorption, wall panels could work as sound insulation panels. However, it is possible that in some cases transferring sound wave pressure would increase, because of a resonance frequency. Some research has already been carried out on sound transmission loss (STL) measurements of acoustic metamaterial wall panels (1.0 m $\times$ 1.2 m) with and without Helmholtz resonators mounted on them. It was observed that additional peaks can be introduced in the STL spectrum, which can be attributed to the Helmholtz resonance of the added resonators [30].

## 5. Conclusions

1. Structures with a longer resonator design absorb sound better at lower frequencies. The longer the structure, the more the absorption peak shifts to low frequencies. For example, a 50 mm long "X"-shaped resonator reaches a sound absorption peak of 0.9 at 800 Hz, while a 100 mm "X"-shaped resonator reaches a sound absorption peak of 0.94 at 315 Hz. This means that different-length resonators should be used in different situations. For example, longer design resonators should be used on the ceiling of a factory or similar buildings with heavy machinery equipment, where low-frequency noise is dominant, while shorter resonators should be integrated into rooms with more user-friendly environments such as schools, shopping malls, and conference halls for shorter reverberation time from day-to-day noise.

2. For the measured plates, it was found that the sound absorption was better with 20 mm thickness plates and with larger hole perforation. With a smaller thickness, there is no significant difference with different perforation layouts or diameters of the holes. The bigger hole perforation allows the sound wave to transmit inside the structure where the resonator containing filler can absorb them. Because of their thickness and the holes in the structure, plates only absorb high frequencies. However, combined with resonators, we can see two absorption peaks. For example, a 50 mm long "O"-shaped resonator used alone has a 0.96 sound absorption peak at 630 Hz frequency, while a 20 mm thick plate with central 2.0 mm perforation used alone has a sound absorption peak of 0.95 at 2500 Hz frequency; while the combined structure with the same 50 mm long "O"-shaped resonator and 20 mm thickness plate with central 2.0 mm perforation has two sound absorption peaks—0.94 at 500 Hz and 0.90 at 1600 Hz. Depending on the situation, the correct combination of perforation and plate thickness together with a suitable resonator should improve room reverberation time and reduce overall noise in the room.

3. The rubber filler percentage should not exceed 50% of a resonator volume to work as intended for sound absorption. The filler position should be at the end of the resonator furthest from the sound source because, when the filler is close to the sound source, it prevents the resonator from fully functioning and absorbing sound waves. Rubber filler provided additional absorption at higher frequencies, which is especially helpful in structures with a longer design resonator. Because of this, we recommend

the use of rubber filler only with longer resonators in cases where there is a need to improve the speech transmission index (STI).

4.  A single-layer panel with the correct perforation and thickness should be able to replace plasterboard as a wall panel material, and a resonator filled with rubber granules should be able to replace building wool. From this research, we can see that this product integration is possible and can be conceptualized even further in the future. By integrating this solution, there should be less waste after demolishing a whole building, or simply renovating part of a building, generating less residual waste and a lower impact on the environment.

**Author Contributions:** Conceptualization, T.J. and A.N.; methodology, T.J.; software, A.N.; validation, A.N.; formal analysis, A.N. and T.J.; investigation, A.N.; resources, T.J.; data curation, T.J.; writing—original draft preparation, A.N.; writing—review and editing, T.J.; visualization, A.N.; supervision, T.J.; project administration, T.J.; funding acquisition, T.J. All authors have read and agreed to the published version of the manuscript.

**Funding:** This research received no external funding.

**Institutional Review Board Statement:** Not applicable.

**Informed Consent Statement:** Not applicable.

**Data Availability Statement:** Research details can be provided upon request to the corresponding author.

**Acknowledgments:** We would like to thank the VILNIUS TECH Institute of Environmental Protection for providing 3D printing materials and access to the 3D printer itself.

**Conflicts of Interest:** The authors declare no conflict of interest.

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
