# Peer review of "Development and Research of Recyclable Composite Metamaterial Structures Made of Plastic and Rubber Waste to Reduce Indoor Noise and Reverberation"

_sustainability, doi:10.3390/su15021731_

Round 1
Reviewer 1 Report
I read the manuscript with high excitement. Even though this manuscript is fixed with the scope of the journal. However, there are some problems for authors to improve their manuscripts before acceptance.
1. The abstract is too long. According to the MDPI guidelines, it should not exceed 200 words. The abstract in the publication exceeds 300 words. It needs to be shortened.
2. Perhaps the authors can add and consult and refer to the following recent references related to waste plastic, as follows the concept of CE. Jamnongkan et al., “Towards a Circular Economy: Study of the Mechanical, Thermal, and Electrical Properties of Recycled Polypropylene and Their Composite Materials”. Polymers 2022, 14, 5482. https://doi.org/10.3390/polym14245482
3. The quality of Figure 1 should be improved? Please add the title name of x- and y-axis.
4. In line 273, what is the coefficient R?. Besides, the symbol “α” in equation 1 should be indicated and explained.
5. Typically, the shape of the material would be affected to the sound absorption. Did you investigate this parameter? If yes, how about the result?
6. Why author should perform the sound absorption at frequencies in the range of 160-5000 Hz.
7. The result and discussion part can be strengthened with more references.
Author Response
Dear reviewer,
Thank you for your time reading my research and for providing comments related to it. I will revise my work according to your comments as clearly as possible.
- Abstract will be shorten;
- I will investigate provided manuscript on plastic waste CE and will add to my research;
- X and Y axis will be added to figure 1;
- α in equation 1 and coefficient R will be explained;
- In my work, two shapes were investigated: X and O shape. The first one were supposed to absorb lower frequencies duo to its spring properties, the second was supposed to absorb higher frequencies to due air particle friction. No other shaped were investigated;
- From work experience (building acoustic measurements) I noticed that this is the typical frequency range for reverberation time classification (as example 500 Hz, 1000 Hz and 2000 Hz for residential house). Lower frequencies are not related to this problem. This is why we picked 30 mm impedance tube, which measurement range is from 160-5000 Hz;
- I will edit and add more reference to discussion part.
Reviewer 2 Report
The manuscript addresses theDevelopment and research of recyclable composite metamaterial constructions made of plastic and rubber waste to reduce indoor noise and reverberation.
The context for the research is clearly stated. The paper covers a suitable research gap. However, the purpose and direct applications of the paper is not clearly stated.
Methodology is not well explained. How are the results analysed? And how do you measure sound absorption?
Results are not compared with literature and there is little critical discussion within the section. Almost no references in the discussion. Paper is too descriptive.
Quality of figures is low and they are not consistent as for formatting.
Reliability of results? Reproducibility? These points must be specified.
Conclusions are not well justified and not related to the main findings. They must be improved.
Author Response
Dear reviewer,
Thank you for your time reading my research and for providing comments related to it. I will revise my work according to your comments as clearly as possible.
- I will try to clearly indicate purpose and direct applications of my research object;
- Sound absorption were measured according to standard with impedance tube using transfer matrix method. Later results were analysed using AcoustiStudio and MatLab software. I will try to add more information and explanations in methodology;
- I will add more references in discussion section. Every metamaterial is unique, perhaps I can compare my material with traditional solutions in construction, like plasterboard, foam or "armstrong" system ceiling;
- I will recheck my figures quality and improve them if possible;
- I will edit or rewrite my conclusion section to relate to the main findings.
Reviewer 3 Report
This paper contains extensive studies and data on the fabrication and mechanism of recyclable composite metamaterial based on plastic and rubber waste to reduce indoor noise and reverberation. The paper is very well written and has values insight that can be useful for the scientific community and industry. The methodology is scientific, and the article is organised and written according to the standards of the journal, with the exception of some minor grammatical errors in the sentence structure. The article lacks sufficient review of recent literature. I suggest a minor revision before considering publication.
Author Response
Dear reviewer,
Thank you for your time reading my research and for providing comments related to it. I will revise my work according to your comments as clearly as possible.
I will double check my grammar in this research before posting revised version again.
Round 2
Reviewer 1 Report
The authors have substantially and satisfactorily revised the manuscript.
Reviewer 2 Report
PAper has significantly improved and I recommend it for publication